# Coordination of cytochrome *bc*₁ complex assembly at MICOS

Ralf M Zerbes[1,2,12], Lilia Colina-Tenorio[1,12], Maria Bohnert [1,3], Karina von der Malsburg[4,5], Christian D Peikert [6,11], Carola S Mehnert[1,2], Inge Perschil[1], Rhena F U Klar[1,7], Rinse de Boer [8], Anita Kram [8], Ida van der Klei[8], Silke Oeljeklaus [9], Bettina Warscheid[6,9], Heike Rampelt [1,10 ✉] & Martin van der Laan [4,5 ✉]

## Abstract

**The boundary and cristae domains of the mitochondrial inner membrane are connected by crista junctions. Most cristae membrane proteins are nuclear-encoded and inserted by the mitochondrial protein import machinery into the inner boundary membrane. Thus, they must overcome the diffusion barrier imposed by crista junctions to reach their final location. Here, we show that respiratory chain complexes and assembly intermediates are physically connected to the mitochondrial contact site and cristae organizing system (MICOS) that is essential for the formation and stability of crista junctions. We identify the inner membrane protein Mar26 (Fmp10) as a determinant in the biogenesis of the cytochrome *bc*₁ complex (complex III). Mar26 couples a Rieske Fe/S protein-containing assembly intermediate to MICOS. Our data indicate that Mar26 maintains an assembly-competent Rip1 pool at crista junctions where complex III maturation likely occurs. MICOS facilitates efficient Rip1 assembly by recruiting complex III assembly intermediates to crista junctions. We propose that MICOS, via interaction with assembly factors such as Mar26, contributes to the spatial and temporal coordination of respiratory chain biogenesis.**

**Keywords** Mitochondria; Cristae; Respiratory Chain; MICOS; *bc*₁ Complex
**Subject Categories** Membranes & Trafficking; Organelles

## Introduction

Conversion of the energy contained in nutrients into the universal cellular energy currency adenosine triphosphate (ATP) is a fundamental metabolic program in all living organisms. In eukaryotic cells the majority of ATP is synthesized in mitochondria via a process termed oxidative phosphorylation (OXPHOS). The protein machineries that carry out the underlying chemical reactions—the respiratory chain complexes and the $F_1F_o$-ATP synthase—are embedded into the mitochondrial inner membrane that is composed of two subcompartments with distinct functions and protein compositions. The inner boundary membrane is in close proximity to the outer membrane and is enriched in transporters, like the components of the mitochondrial protein import machinery (Zick et al, 2009; Horvath et al, 2015). The oxidative phosphorylation system of mitochondria is mainly localized to the cristae of the inner membrane that shape a specialized microcompartment for chemi-osmotic coupling (Gilkerson et al, 2003; Mannella, 2006; Vogel et al, 2006; Wurm and Jakobs, 2006; Zick et al, 2009; Davies et al, 2011; Appelhans et al, 2012; Wilkens et al, 2013; Cogliati et al, 2016; van der Laan et al, 2016; Kondadi et al, 2020; Colina-Tenorio et al, 2020; Kondadi and Reichert, 2024). Cristae membranes are highly folded invaginations protruding from the inner boundary membrane into the central matrix compartment. They are connected to the boundary membrane via narrow tubular openings, the crista junctions, that are thought to act as a diffusion barrier (Perkins et al, 1997; Frey et al, 2002; Mannella, 2006; Zick et al, 2009; Wollweber et al, 2017; Wolf et al, 2019; Colina-Tenorio et al, 2020). The asymmetric protein distribution between inner boundary and cristae membranes imposes a logistical challenge for the mitochondrial protein sorting system. Many of the membrane-integral subunits of the OXPHOS machinery are encoded by nuclear genes and synthesized in the cytosol as cleavable precursor proteins with an aminoterminal presequence. These proteins are recognized by dedicated mitochondrial surface receptors and enter the organelle via the general protein translocase of the outer membrane (TOM complex) (Harbauer et al, 2014; Araiso et al, 2022; Busch et al, 2023). Insertion into the inner boundary membrane is mediated by the

[1]Institute of Biochemistry and Molecular Biology, ZBMZ, Faculty of Medicine, University of Freiburg, 79104 Freiburg, Germany. [2]Faculty of Biology, University of Freiburg, 79104 Freiburg, Germany. [3]Institute of Cell Dynamics and Imaging, Cells in Motion Interfaculty Centre (CiM), University of Münster, 48149 Münster, Germany. [4]Medical Biochemistry and Molecular Biology, Saarland University, 66421 Homburg, Germany. [5]Center for Molecular Signaling, PZMS, Saarland University, 66421 Homburg, Germany. [6]BIOSS Centre for Biological Signalling Studies, Universität Freiburg, 79104 Freiburg, Germany. [7]Institute of Molecular Medicine and Cell Research (IMMZ), University of Freiburg, 79104 Freiburg, Germany. [8]Molecular Cell Biology, University of Groningen, 9700 CC Groningen, The Netherlands. [9]Faculty of Chemistry and Pharmacy, Biochemistry II, Theodor Boveri-Institute, University of Würzburg, 97074 Würzburg, Germany. [10]CIBSS Centre for Integrative Biological Signalling Studies, University of Freiburg, 79104 Freiburg, Germany. [11]Present address: Bioinformatics Research & Development, BioNTech SE, 55131 Mainz, Germany. [12]These authors contributed equally: Ralf M Zerbes, Lilia Colina-Tenorio. ✉E-mail: heike.rampelt@biochemie.uni-freiburg.de; martin.van-der-laan@uks.eu

presequence translocase of the inner mitochondrial membrane (TIM23 complex) and its partner protein complexes (Mokranjac and Neupert, 2010; Schulz et al, 2015; Moulin et al, 2019; Busch et al, 2023; Fielden et al, 2023; Zhou et al, 2023). Thus, somewhere on the way from membrane insertion of OXPHOS proteins at the inner boundary membrane to their final destination in the cristae membranes, the diffusion barrier imposed by the crista junctions must be crossed. In fact, early versus late assembly steps of complex III and IV are localized asymmetrically: While early steps preferentially take place in the inner boundary membrane, late ones are more prevalent in the cristae membranes (Stoldt et al, 2018).

Crista junctions with their high local membrane curvature require for their stability the mitochondrial contact site and cristae organizing system (MICOS) (Harner et al, 2011; Hoppins et al, 2011; von der Malsburg et al, 2011; Friedman and Nunnari, 2014; Kozjak-Pavlovic, 2017; Rampelt et al, 2017; Wollweber et al, 2017). The MICOS complex is highly conserved in evolution and consists of at least six different genuine subunits in yeast and seven in mammals that are organized in two distinct modules (Rabl et al, 2009; Harner et al, 2011; Hoppins et al, 2011; von der Malsburg et al, 2011; Alkhaja et al, 2012; Ott et al, 2012; Pfanner et al, 2014; Guarani et al, 2015; Muñoz-Gómez et al, 2015; Huynen et al, 2016; Colina-Tenorio et al, 2020; Mukherjee et al, 2021; Bock-Bierbaum et al, 2022). One subcomplex consist of Mic60, Mic19, and in mammals additionally Mic25, and forms contact sites between inner and outer mitochondrial membranes through multiple interactions with outer membrane protein complexes. Moreover, Mic60 induces membrane curvature via an amphipathic helix within its intermembrane space domain (Hessenberger et al, 2017; Tarasenko et al, 2017). The other subcomplex is composed of large oligomers of Mic10, a small inner membrane protein with an intrinsic membrane-bending activity, together with Mic12/QIL1, Mic26 and Mic27 (Barbot et al, 2015; Bohnert et al, 2015; Friedman et al, 2015; Guarani et al, 2015). Both MICOS subcomplexes are necessary for the formation of crista junctions, and their physical coupling is largely mediated by Mic12/QIL1 (Guarani et al, 2015; Zerbes et al, 2016). MICOS deficiency causes the loss of crista junctions and the detachment of cristae from the inner boundary membrane, and loss of function mutations in human patients cause severe mitochondrial pathologies, including a fatal encephalopathy (John et al, 2005; Rabl et al, 2009; Mun et al, 2010; Harner et al, 2011; Hoppins et al, 2011; von der Malsburg et al, 2011; Guarani et al, 2016; Zeharia et al, 2016; Benincá et al, 2021; Peifer-Weiß et al, 2023).

Respiratory chain biogenesis is a highly complicated multi-step process that requires a plethora of dedicated assembly factors. Moreover, individual respiratory chain complexes associate to form supercomplexes of different stoichiometry (Enríquez, 2016; Hartley et al, 2019; Rathore et al, 2019; Vercellino and Sazanov, 2022). Whereas major assembly steps and intermediates in the biogenesis of NADH dehydrogenase (complex I) and cytochrome $c$ oxidase (complex IV) have been identified and characterized (Mick et al, 2011; Soto et al, 2012; Stroud et al, 2016; Guerrero-Castillo et al, 2017; Formosa et al, 2018; Timón-Gómez et al, 2018), comparably little is known about the mechanism of cytochrome $bc_1$ complex (complex III) assembly. In the yeast *Saccharomyces cerevisiae*, complex III is composed of ten different subunits (Smith et al, 2012; Ndi et al, 2018; Signes and Fernandez-Vizarra, 2018; Zara

et al, 2022) and forms dimeric supercomplexes that are found associated with either one or two copies of complex IV (Wittig and Schägger, 2009; Hartley et al, 2019; Rathore et al, 2019). Complex III assembly is initiated by the translation and membrane insertion of the mitochondrially encoded subunit cytochrome $b$ (Cob), which subsequently forms an early core subcomplex together with Qcr7 and Qcr8 (Zara et al, 2007; Gruschke et al, 2011; 2012). A late complex III assembly intermediate of about 500 kDa was identified that is already dimeric and contains all subunits except the Rieske Fe/S protein (Rip1) and Qcr10 (Zara et al, 2009; Conte et al, 2015; Stephan and Ott, 2020). Incorporation of these two proteins and formation of supercomplexes with complex IV constitute the final steps in complex III assembly (Cruciat et al, 1999; Wagener et al, 2011; Atkinson et al, 2011; Cui et al, 2012; Smith et al, 2012; Ndi et al, 2018; Kater et al, 2020; Tang et al, 2020).

Accumulating evidence suggests that mitochondrial membrane architecture and respiratory chain integrity are closely linked. Alterations of cristae morphology lead to defects in respiratory chain supercomplex formation and decreased respiratory capacity (Cogliati et al, 2013, 2016; Baker et al, 2019; Colina-Tenorio et al, 2020). In cells with defective MICOS complexes, mitochondrial respiration is considerably reduced and the distribution of respiratory chain complexes in the inner mitochondrial membrane appears to be altered (von der Malsburg et al, 2011; Weber et al, 2013; Harner et al, 2014; Bohnert et al, 2015; Friedman et al, 2015; Guarani et al, 2015; Anand et al, 2020; Rampelt et al, 2022). However, the molecular nature of the interaction network that links respiratory chain biogenesis to cristae formation and remodeling has remained enigmatic.

Here we show that MICOS physically associates with respiratory chain (super-)complexes and distinct respiratory chain assembly intermediates. We have identified the so far uncharacterized inner mitochondrial membrane protein Mar26 as an interaction partner of both respiratory chain complexes and MICOS. Mar26 is part of a novel Rip1-containing complex III assembly intermediate and couples this subcomplex to MICOS via the Mic60-Mic19 module. Loss of Mar26 leads to decreased respiratory growth and perturbs late stages of complex III biogenesis, indicating that Mar26 directly contributes to late complex III assembly steps. Recruitment of the Rip1 assembly intermediate to MICOS at crista junctions facilitates faithful delivery of Rip1 to complex III.

# Results

## MICOS is connected to the respiratory chain via the Mic60-Mic19 module

In a previous study, we determined the interactome of the MICOS core component Mic60 (von der Malsburg et al, 2011). In addition to the five MICOS components Mic10, Mic12, Mic19, Mic26, Mic27 and subunits of the TOM complex, our analysis identified subunits of the respiratory chain as potential interaction partners of Mic60. To investigate the relationship between MICOS and the respiratory chain in more detail, we purified native protein complexes by affinity chromatography via protein A tags on either Mic60 or Mic12 after solubilization of mitochondrial membranes with the mild detergent digitonin. Western blot analysis of the elution fractions revealed a specific co-isolation of core subunits of

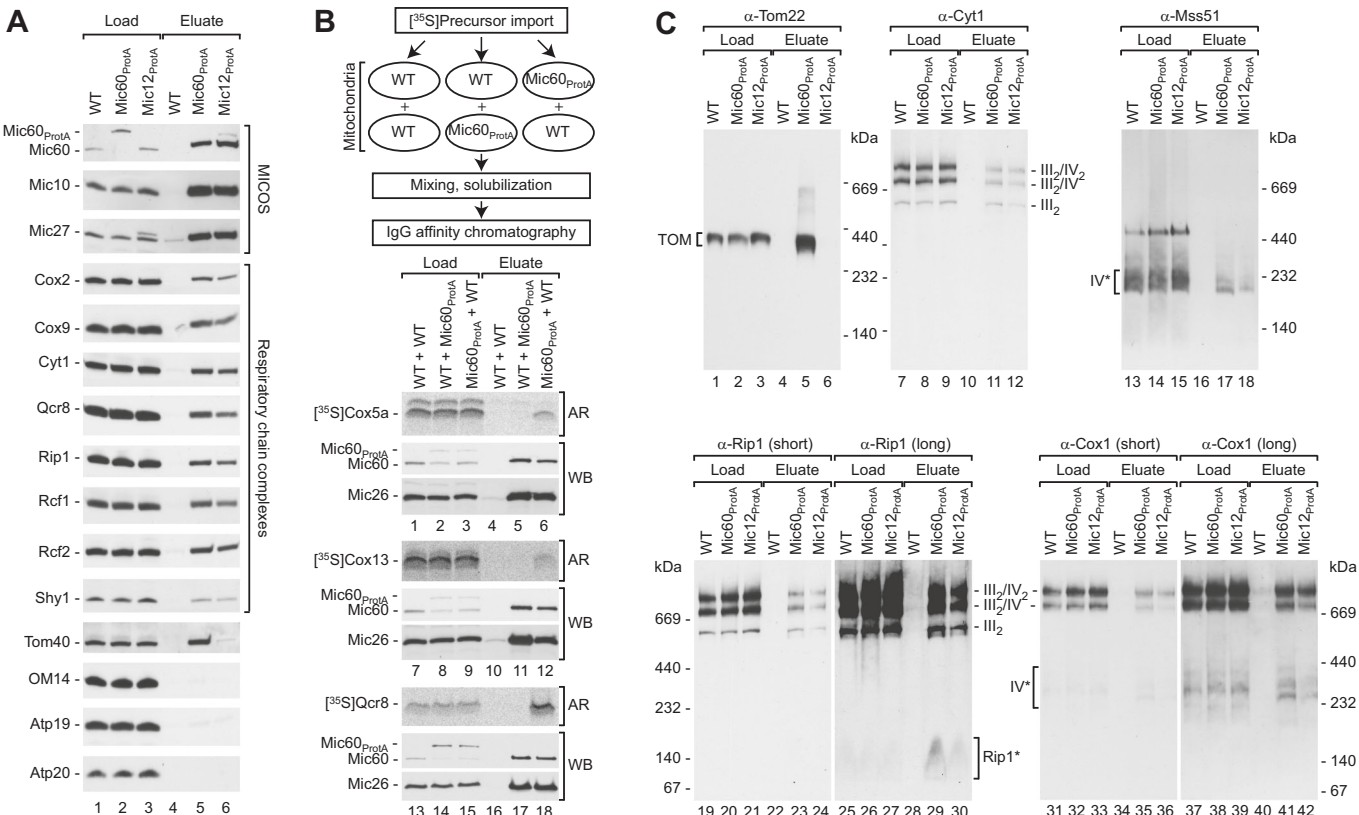

**Figure 1. MICOS is connected to the respiratory chain.**

(A) Protein complexes were purified from digitonin-solubilized wild-type (WT), Mic12ProtA and Mic60ProtA mitochondria by IgG chromatography and analyzed by SDS-PAGE and western blotting. Load, 1%; Eluate, 100%. Mic60, Mic10, Mic12, Mic27, MICOS subunits; Cox2, Cox9, complex IV subunits; Cyt1, Qcr8, Rip1, complex III subunits; Rcf1, Rcf2, Shy1, respiratory chain assembly factors; Tom40, subunit of the translocase of the outer membrane (TOM); OM14, outer membrane protein; Atp19, Atp20, F1Fo-ATP synthase subunits. (B) Indicated [35S]-labeled preproteins were imported either into wild-type (WT) or Mic60ProtA mitochondria. Upon re-isolation mitochondria were mixed with WT or Mic60ProtA mitochondria as indicated. Protein complexes were then purified by IgG chromatography as in (A) and analyzed by SDS-PAGE and western blot (WB) or autoradiography (AR). Load, 1%; Eluate, 100%. Cox5a, Cox13, complex IV subunits; Qcr8, complex III subunit, Mic26, MICOS subunit. (C) Protein complexes were purified as in (A) and analyzed by blue native (BN-)PAGE and immunoblotting. Short and long exposures of blots probed with Rip1 and Cox1 antibodies are shown. Load, 0.5%; Eluate, 100%. III2/IV2, III2/IV, III2, supercomplexes formed by respiratory chain complexes III and IV; IV*, complex IV and assembly intermediates thereof; Cyt1, Rip1, complex III subunits; Rip1*, Rip1-containing assembly intermediate; Mss51, complex IV assembly factor; Cox1, complex IV subunit; Tom22, TOM subunit. Source data are available online for this figure.

respiratory chain complex III (Cyt1, Qcr8, Rip1) and complex IV (Cox2, Cox9) with both Mic60ProtA and Mic12ProtA (Fig. 1A, lanes 4–6). Furthermore, we recovered in the elution fractions the respiratory chain supercomplex-associated proteins Rcf1 and Rcf2 (Strogolova et al, 2012; Vukotic et al, 2012; Chen et al, 2012; Strogolova et al, 2019), as well as the complex IV assembly factor Shy1 (Mick et al, 2007). As reported previously, Tom40 was efficiently co-isolated only with Mic60ProtA, indicating the presence of different Mic60 pools in the inner mitochondrial membrane (von der Malsburg et al, 2011; Bohnert et al, 2012).

To exclude the possibility that MICOS and the respiratory chain components interacted only after detergent-mediated lysis of mitochondria, we performed a post-lysis control experiment: The in vitro synthesized and radiolabeled precursors of Cox5a, Cox13, and Qcr8 were imported either into wild-type or Mic60ProtA mitochondria. After re-isolation, preprotein-loaded wild-type mitochondria were mixed with untreated Mic60ProtA mitochondria and vice versa (Fig. 1B, flow diagram). Mitochondrial samples were then solubilized with digitonin and MICOS complexes were

isolated. Radiolabeled respiratory chain subunits were only co-isolated with tagged Mic60 when they had been imported into Mic60ProtA mitochondria, whereas intrinsic (unlabeled) Mic26 and Mic60 were co-isolated in all cases (Fig. 1B, lanes 4–6, 10–12, and 16–18). We conclude that respiratory chain complexes associate with MICOS within the mitochondrial inner membrane and not after lysis. Blue native PAGE analysis of the import reactions confirmed that newly imported Cox13 and Qcr8 were assembled into respiratory chain supercomplexes (Fig. EV1A, lanes 4–9), whereas the majority of newly imported Cox5a accumulated in complex IV assembly intermediates as previously reported (Fig. EV1A, lanes 1–3) (Mick et al, 2007).

These data suggested that both respiratory chain supercomplexes and assembly factors interact with MICOS. We therefore analyzed the elution fractions of MICOS isolations using tagged Mic60 and Mic12 by blue native PAGE and western blotting. In line with the SDS-PAGE analysis (Fig. 1A), TOM complexes were specifically purified with tagged Mic60, but not with Mic12 as expected (Fig. 1C, lanes 4–6). The elution fractions of both

Mic60$_{ProtA}$ and Mic12$_{ProtA}$ isolations contained considerable amounts of different respiratory chain supercomplex species composed of complex III and IV, demonstrating that mature respiratory chain supercomplexes interact with MICOS (Fig. 1C, lanes 10–12, 22–24, and 34–36). Other abundant proteins such as OM14 of the outer membrane, or inner membrane proteins including the ATP synthase subunits Atp19, Atp20 or Atp21, and the i-AAA protease Yme1 did not interact with MICOS (Figs. 1A and EV1B), confirming the specificity of the observed MICOS interactions with the respiratory chain. Distinct Cox1-containing assembly intermediates of complex IV have been identified that are associated with specialized assembly factors, like Mss51, not present in mature supercomplexes (Mick et al, 2011). These early assembly intermediates were also co-isolated with Mic60$_{ProtA}$ and Mic12$_{ProtA}$ (Fig. 1C, lanes 16–18 and 40–42). Interestingly, using antibodies against Rip1 we found in the elution fractions a small Rip1-containing protein complex that has not been described previously. This complex is barely detectable and thus of low abundance in total mitochondrial extracts, but strongly enriched with purified MICOS complexes (Fig. 1C, lanes 22–24 and 28–30). Hence, the small Rip1-containing complex efficiently associates with MICOS and potentially represents a complex III assembly intermediate.

The MICOS complex is composed of two distinct subcomplexes that both contribute to membrane curvature generation at crista junctions: a Mic10–Mic12–Mic26–Mic27 module and a Mic60-Mic19 module that interacts with outer mitochondrial membrane protein complexes to form membrane contact sites (Barbot et al, 2015; Bohnert et al, 2015; Friedman et al, 2015; Hessenberger et al, 2017; Tarasenko et al, 2017; Bock-Bierbaum et al, 2022). To investigate which MICOS module is responsible for respiratory chain coupling, affinity chromatography and blue native PAGE analysis were performed with Mic60$_{ProtA}$ *mic10Δ* mitochondria. Deletion of *MIC10* leads to MICOS disruption and loss of crista junctions (Harner et al, 2011; Hoppins et al, 2011; von der Malsburg et al, 2011; Bohnert et al, 2015). Loss of Mic10 did not abolish the co-isolation of respiratory chain complexes III and IV with tagged Mic60 (Fig. EV1C, lanes 4–6, 10–12, 16–18; Fig. EV1E). These data indicate that respiratory chain complexes are coupled to MICOS via the Mic60-Mic19 module and that the Mic10-containing module is not required for the respiratory chain interaction of Mic60-Mic19. In line with this conclusion, the amounts of respiratory chain complexes co-purified with Mic12$_{ProtA}$ in the absence of Mic60 or Mic10 were only slightly above the unspecific background (Fig. EV1D, lanes 5–8, 13–16). In Mic10-deficient mitochondria, Mic60 is still present but not associated with the Mic12-containing MICOS subcomplex (Bohnert et al, 2015). We conclude that the Mic60-Mic19 module recruits respiratory chain complexes to MICOS.

## Mar26 is associated with MICOS and respiratory chain complexes

To further analyze the connection between the respiratory chain and MICOS complexes, we used yeast cells expressing TAP-tagged Cor1 of complex III (van der Laan et al, 2006). Stable isotope labeling with amino acids in cell culture (SILAC) combined with high-resolution mass spectrometry (MS) (von der Malsburg et al, 2011) was used to identify respiratory chain-interacting proteins (Dataset EV1). Besides the genuine subunits of respiratory chain

complexes III and IV, we co-isolated components of the presequence translocase of the inner mitochondrial membrane and a number of metabolite carrier proteins as previously reported (van der Laan et al, 2006; Claypool et al, 2008; Dienhart and Stuart, 2008; Mehnert et al, 2014) (Dataset EV1; Fig. 2A). Moreover, MICOS components were also significantly enriched with tagged respiratory chain complexes, independently confirming the physical association of these protein machineries (Fig. 2A). Interestingly, we also found a strong enrichment of a so far uncharacterized mitochondrial protein encoded by the *S. cerevisiae* open reading frame *YER182W* that was named Found in mitochondrial proteome 10 (Fmp10). We will below refer to this protein as Mar26 (MICOS-associated respiratory chain factor of 26 kDa). Our SILAC/MS-based analysis of the Mic60 interactome had previously identified Mar26 as a putative partner protein of MICOS (von der Malsburg et al, 2011). (For comparison, we have included a new analysis of the original Mic60$_{ProtA}$ SILAC/MS dataset performed as for Cor1$_{TAP}$ isolations in Dataset EV1.)

We generated an antiserum against a C-terminal peptide of Mar26. Western blot analysis of Cor1$_{TAP}$ and Mic60$_{ProtA}$ isolations confirmed that Mar26 is efficiently co-isolated with both respiratory chain complexes and MICOS (Fig. 2B,C). We asked if the presence of a functional respiratory chain is necessary for the stable expression and accumulation of Mar26 in mitochondria. We generated a Mic60$_{ProtA}$ yeast strain lacking mitochondrial DNA (*rho⁻*). Respiratory chain complex subunits that are encoded by nuclear genes, like Rip1, Qcr8, Cox4, or Cox9, are still detectable in isolated mitochondria of *rho⁻* cells (Fig. EV2A) but cannot assemble into functional respiratory chain complexes due to the lack of the mitochondrial encoded subunits. Therefore, their levels are considerably reduced compared to *rho⁺* mitochondria. In contrast, the steady-state levels of Mar26 as well as the outer membrane preprotein receptor Tom70 were comparable in *rho⁺* and *rho⁻* mitochondria, and those of the MICOS core components Mic60 and Mic10 were only mildly reduced (Fig. EV2A). We conclude that accumulation of Mar26 in mitochondria does not depend on the presence of respiratory chain complexes. Affinity chromatography experiments furthermore revealed that Mar26 is still efficiently co-isolated with tagged Mic60 in the *rho⁻* mitochondria lacking respiratory chain complexes (Fig. EV2B).

Homology searches revealed that *MAR26* has a paralog in *S. cerevisiae* encoded by the open reading frame *YBL095W*. Like Mar26, the encoded protein was found associated with mitochondrial ribosomes and therefore named Mrx3 (Kehrein et al, 2015). It was identified as a potential MICOS-interacting protein (von der Malsburg et al, 2011; Jin et al, 2015), and found enriched in our Cor1$_{TAP}$ isolations (Fig. 2A). Mar26 and Mrx3 share 49.3% similarity and 16.4% identity (Fig. EV2C). We did not succeed in generating antibodies against Mrx3 and instead fused a protein A tag to the protein for biochemical analysis. Alkaline extraction of mitochondrial membranes showed that both Mar26 and Mrx3$_{ProtA}$ remained in the pellet fraction together with other integral membrane proteins, like Tom70 or Tim23, at pH 10.8 (Fig. 2D). Mar26 remained in the pellet also at pH 11.5, whereas Mrx3 was found in the supernatant fraction at pH 11.5 (Fig. 2D). This behavior is in agreement with the predicted presence of one transmembrane segment in Mar26 and the low predicted hydrophobicity of a potential Mrx3 transmembrane segment. The submitochondrial localization of both proteins was tested by a

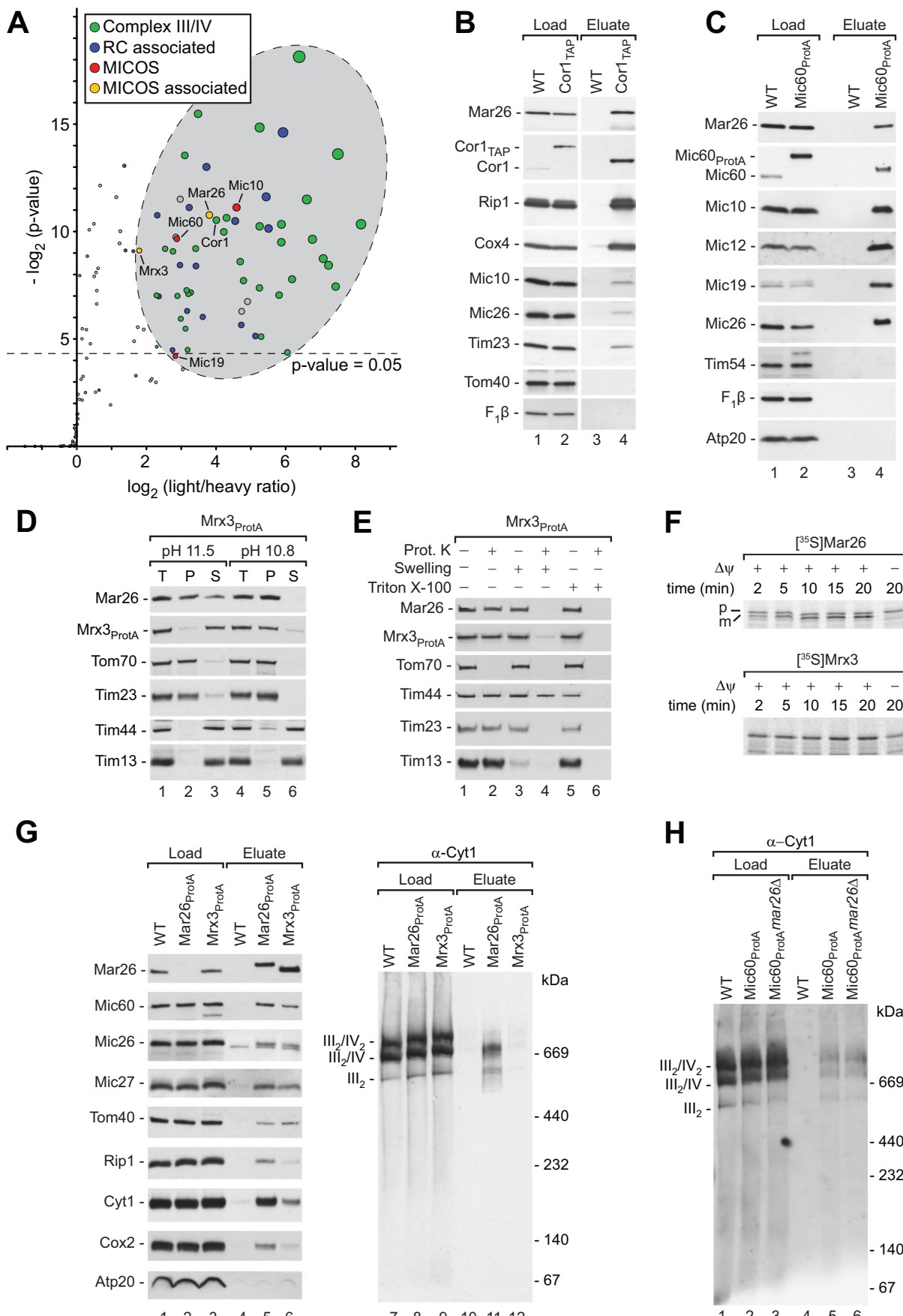

◀

**Figure 2. Interaction network of Mar26, MICOS and respiratory chain complexes.**

(A) SILAC-based MS analysis of Cor1$_{TAP}$ interactors. Structural subunits of complex III and IV and known proteins required for complex III and IV biogenesis (green), proteins described previously to associate with respiratory chain (RC) complexes III and IV (blue), MICOS components (red) and newly identified MICOS-associated respiratory chain factors (yellow) are indicated. Additional proteins labeled in gray have not been reported before to associate with the respiratory chain (Rad4, Tao3, Ynl122c). A one-sided $t$ test was performed to determine the $P$ value for each protein. A complete list of identified proteins can be found in Dataset EV1. Cor1, complex III subunit. (B) Protein complexes were purified from wild-type (WT) and Cor1$_{TAP}$ mitochondria by IgG chromatography and analyzed by SDS-PAGE and western blotting. Load, 4%; Eluate, 100%. Tim23, subunit of the TIM23 presequence translocase. F$_1$β, β subunit of F$_1$F$_o$-ATP synthase. (C) Protein complexes purified from wild-type and Mic60$_{ProtA}$ mitochondria were analyzed as in (B). Load, 4%; Eluate, 100%. Mic19, MICOS subunit; Tim54, subunit of the TIM22 carrier translocase. (D) Membrane association of Mar26 und Mrx3 was tested by alkaline extraction and subsequent ultracentrifugation. T total, P pellet, S supernatant. Tom70, TOM component; Tim44, TIM23 component; Tim13, small Tim chaperone. (E) Submitochondrial localization of Mar26 and Mrx3 was assessed by protease K accessibility. Mrx3$_{ProtA}$ mitochondria were either incubated in SEM buffer (intact mitochondria), subjected to hypoosmotic swelling or solubilized with 0.5% (v/v) Triton X-100 and subsequently treated with proteinase K (Prot. K) as indicated. (F) [$^{35}$S]Mar26 and [$^{35}$S]Mrx3 preproteins were imported into WT mitochondria in the presence or absence of an inner membrane potential (Δψ). Non-imported preproteins were removed by proteinase K treatment. Samples were analyzed by SDS-PAGE and autoradiography. In vitro synthesized preprotein (Lysate) is shown for comparison where indicated. p precursor, m mature protein. (G) Protein complexes of wild-type (WT), Mar26$_{ProtA}$ and Mrx3$_{ProtA}$ were purified under native conditions and analyzed by SDS-PAGE or BN-PAGE and western blotting. Load, 1%; Eluate, 100%. III$_2$/IV$_2$, III$_2$/IV, III$_2$, supercomplexes formed by respiratory chain complexes III and IV. (H) Analysis of respiratory chain supercomplexes co-purified with Mic60$_{ProtA}$ in the presence or absence of Mar26 as indicated by BN-PAGE and western blotting. Load, 1%; Eluate, 100%. Source data are available online for this figure.

protease accessibility assay (Fig. 2E). When proteinase K is added to intact mitochondria, only surface-exposed outer membrane proteins, like Tom70, are digested. Hypoosmotic swelling opens up the outer membrane and allows access of the protease to proteins that expose soluble domains to the intermembrane space, like Tim23 or Tim13. Under these conditions, also Mar26 and Mrx3$_{ProtA}$ were proteolytically degraded. Matrix-exposed proteins, like Tim44, remained protected and were only degraded upon detergent solubilization of mitochondria (Fig. 2E). We conclude that Mar26 and Mrx3 are both inner membrane proteins exposed to the intermembrane space.

No presequence cleavage site is predicted for Mrx3. In agreement with this prediction, the in vitro synthesized and radiolabeled Mrx3 preprotein was imported into isolated mitochondria to a protease-protected localization in a membrane-potential-dependent manner, but not proteolytically processed. Radiolabeled Mar26 was likewise imported into mitochondria, but processed to a mature form indicative of presequence cleavage (Fig. 2F). This is consistent with the finding that mature Mar26 is generated by the successive processing of $12 + 1$ N-terminal residues by the presequence peptidase MPP and the intermediate cleaving peptidase Icp55, respectively (Vögtle et al, 2009).

## Mar26, MICOS and the respiratory chain form an interaction network

We performed Mar26 and Mrx3 affinity purifications using Protein A-tagged variants of these proteins expressed from their native chromosomal loci. Initial SILAC-MS analysis of the Mar26 interactome revealed a strong interaction with Mrx3 and confirmed its close association with MICOS and respiratory chain supercomplexes (Dataset EV1). Further western blot analysis showed that Mar26$_{ProtA}$ and Mrx3$_{ProtA}$ co-purified MICOS components as well as the central TOM complex subunit Tom40 with similar efficiency (Fig. 2G, lanes 4–6). MICOS components were recovered together with Mar26$_{ProtA}$ and Mrx3$_{ProtA}$ also from $rho^-$ mitochondria showing that the association of both Mar26 and Mrx3 with MICOS does not depend on the presence of respiratory chain complexes (Fig. EV2D). Using $rho^+$ mitochondria, subunits of complex III, like Rip1 or Cyt1, and of complex IV, like Cox2, were considerably more abundant in the elution fractions of Mar26$_{ProtA}$

isolations compared to Mrx3$_{ProtA}$ (Fig. 2G, lanes 4–6). Accordingly, mature respiratory chain supercomplexes, as detected with antibodies against the complex III subunit Cyt1, were co-isolated with Mar26$_{ProtA}$, but only in minor amounts with Mrx3$_{ProtA}$ (Fig. 2G, lanes 10–12). Thus, Mar26 and Mrx3 are both MICOS-binding proteins, but the connection to the respiratory chain is clearly more evident for Mar26.

We next tested whether Mar26 may be involved in the recruitment of respiratory chain supercomplexes to MICOS. However, affinity purification experiments with mitochondria from cells expressing Mic60$_{ProtA}$ in the absence of Mar26 revealed that the recovery of mature respiratory chain supercomplexes in the elution fractions was similar for Mic60$_{ProtA}$ and Mic60$_{ProtA}$ $mar26\Delta$ mitochondria (Fig. 2H, lanes 4–6). Thus, Mar26 is not required for coupling of mature respiratory chain supercomplexes to MICOS.

## Lack of Mar26 impairs respiratory growth and complex III activity

What is the functional role of Mar26 in mitochondria? To find out whether Mar26 is important for respiratory metabolism, we tested the growth of the deletion mutant at elevated temperature on fermentable and respiratory medium containing either glucose or ethanol and glycerol as carbon sources. Notably, $mar26\Delta$ cells showed a substantial growth defect specifically on solid respiratory medium (Fig. 3A). We additionally assayed growth in defined liquid media using an approach that allows us to monitor fermentation, metabolic adaptation and respiration in the same assay by including glycerol and a small amount of glucose (Rampelt et al, 2022). During the initial fermentation phase, lack of Mar26 did not result in any defect; however, $mar26\Delta$ cells displayed a severe growth delay during both metabolic adaptation and respiration, specifically at elevated temperature (Fig. 3B). Based on our interactome analysis (Fig. 2A; Dataset EV1), we first asked whether cristae architecture was affected in these mutants and consequently examined mutant mitochondria by electron microscopy. Mitochondrial ultrastructure, cristae morphology and the prevalence of crista junctions were indistinguishable from the corresponding wild-type indicating that MICOS function is not considerably impaired in the absence of Mar26 (Fig. 3C). Given the impaired respiratory growth of Mar26-deficient mutant cells, we

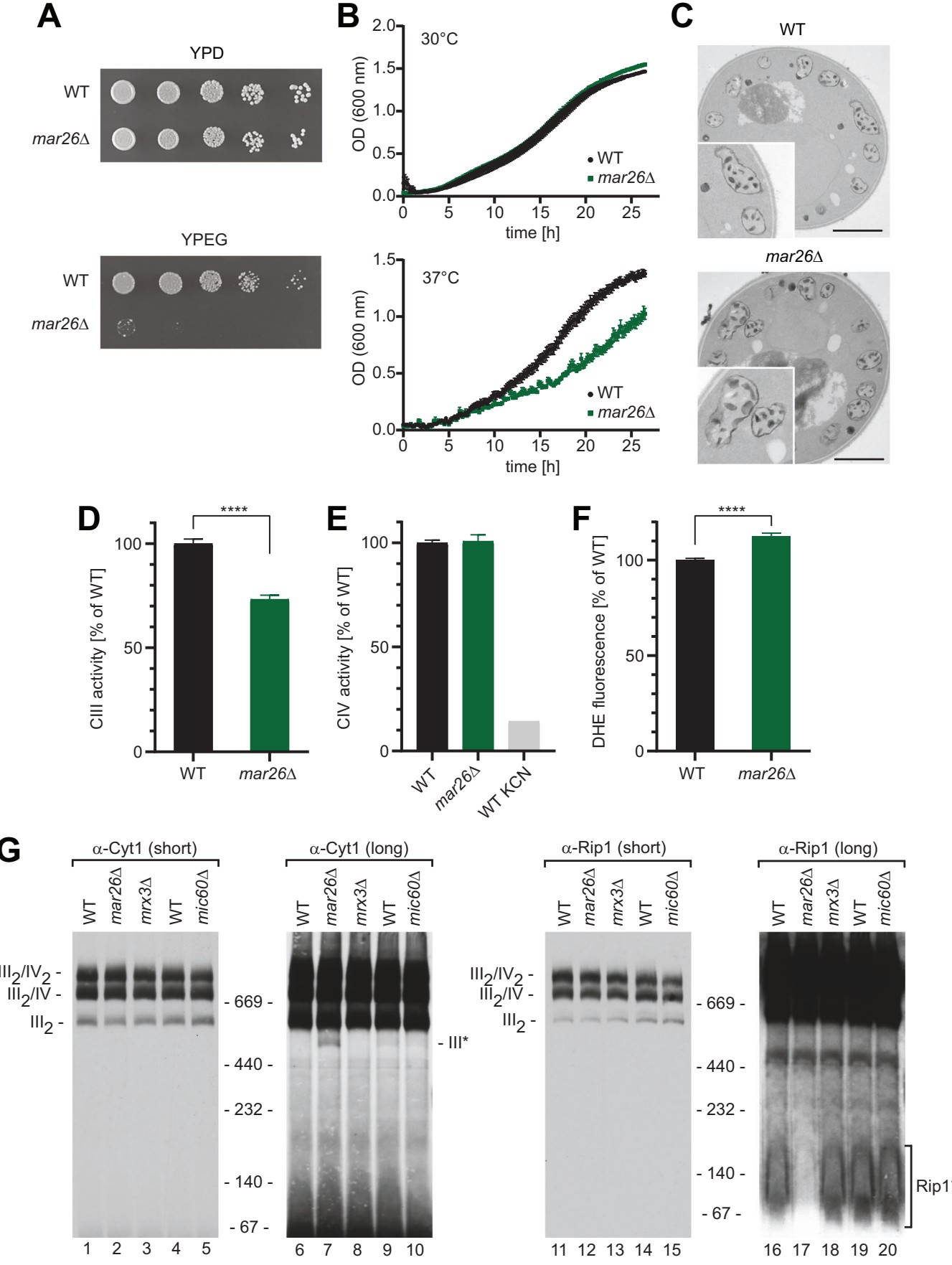

**Figure 3. Mar26 deficiency leads to a growth defect and decreased complex III activity.**

(A) WT and *mar26Δ* cells were spotted as fivefold serial dilutions on YPD or YPEG plates and incubated at 34 °C. (B) WT and *mar26Δ* cells were grown in liquid synthetic defined medium with 3% glycerol and 0.1% glucose as carbon sources. Upper panel: 30 °C; lower panel: 37 °C. Error bars: SEM; $n = 8$ (2 independent experiments). (C) Electron microscopy analysis of wild-type (WT) and *mar26Δ* cells using diaminobenzidine staining of mitochondrial cristae. Bar, 1 μm. (D) Complex III activity was assessed photometrically by measuring reduction of cytochrome $c$ in isolated WT and *mar26Δ* mitochondria. Error bars: SEM; $n = 45 / 41$ (4 independent experiments); $P = 8.1*10^{-14}$ (unpaired two-tailed $t$ test). (E) Complex IV activity was assessed photometrically by measuring oxidation of cytochrome $c$ in solubilized WT and *mar26Δ* mitochondria. Error bars: SEM; $n = 20$ (4 independent experiments). (F) Superoxide formation in WT and *mar26Δ* mitochondria isolated after growth at 37 °C was measured as increase in dihydroethidium (DHE) fluorescence in the presence of the respiratory substrate succinate. Error bars: SEM; $n = 70$ (12 independent experiments); $P = 4.9*10^{-9}$ (unpaired two-tailed $t$ test with Welch correction). (G) Respiratory chain supercomplexes in WT, *mar26Δ*, *mrx3Δ* and *mic60Δ* mitochondria were analyzed by BN-PAGE and immunoblotting. $III_2/IV_2$, $III_2/IV$, $III_2$, supercomplexes formed by respiratory chain complexes III and IV; III*, 500 kDa complex III assembly intermediate; Rip1*, Rip1-containing assembly intermediate. Source data are available online for this figure.

measured the enzymatic activities of the respiratory chain complexes III and IV and $F_1F_o$-ATP synthase (complex V) individually. Of note, complex III activity was considerably lower in *mar26Δ* mitochondria, whereas complexes IV and V were unaffected (Figs. 3E and EV3A). We also observed increased reactive oxygen production in the absence of Mar26 in vivo (Fig. EV3B) as well as in isolated mitochondria in the presence of respiratory substrate (Fig. 3F). The steady-state levels of MICOS subunits and many other mitochondrial proteins tested were normal in *mar26Δ* mitochondria (Fig. EV3C). Respiratory chain supercomplexes and Mss51-containing complex IV assembly intermediates were similarly abundant in wild-type, *mar26Δ*, and *mic60Δ* mutant mitochondria (Fig. 3G, lanes 1–5 and 11–15; Fig. EV3D). In line with the relatively weak association of Mrx3 with the respiratory chain (Fig. 2G), we did not observe alterations in any of the oxidative phosphorylation complexes in the absence of Mrx3 (Figs. 3G and EV3D). In contrast, longer exposures of blue native PAGE western blots probed with antibodies against Cyt1 of complex III revealed the accumulation of a ~500 kDa Cyt1-containing complex in the absence of Mar26 (Fig. 3G, lanes 6–10). The apparent molecular weight of this protein complex resembles that of the 500 kDa late assembly intermediate of complex III that accumulates in the absence of Rip1 (Fig. 3G, lane 7) (Zara et al, 2009). Strikingly, the small complex containing Rip1 that we found enriched with purified MICOS complexes (Fig. 1C) was not detectable in the *mar26Δ* mutant mitochondria, indicating that Mar26 is likely part of this small Rip1 complex (Fig. 3G, lanes 16–17). Although this complex efficiently binds to MICOS via the Mic60-Mic19 module, its formation and stability does not require Mic60, because it is present at normal levels in *mic60Δ* mitochondria (Fig. 3G, lanes 19–20).

## Mar26 plays a regulatory role in complex III biogenesis

Using two-dimensional (native / non-native) gel analysis, we detected a pool of Rip1 molecules that co-migrated with Mar26 in the native dimension (Fig. 4A). Together with the Mar26-dependence of the small Rip1 complex and the reduced complex III activity in Mar26-deficient cells (Fig. 3D), these data further support the idea that Mar26 is involved in Rip1 assembly with complex III. To get more detailed insights, we monitored Mar26 complexes via both native and two-dimensional (native/non-native) gel electrophoresis in mitochondria lacking subunits or biogenesis factors for complex III. In wild-type mitochondria, Mar26 migrated mostly in the low molecular weight range, with only traces found at respiratory chain supercomplexes (Fig. 4B, lane

1–2). In the absence of Rip1, in agreement with the literature (Conte et al, 2015), we observed that the 500 kDa complex, which is a dimer of partially assembled complex III (Stephan and Ott, 2020), already associated with complex IV to form (non-functional) respiratory chain supercomplexes (Fig. 4B, lanes 3 and 9). Strikingly, Mar26 bound to these immature complex III species with higher efficiency than to the wild-type (super-)complexes (Fig. 4B, lanes 4 and 6). A Mar26$_{ProtA}$ affinity purification from wild-type or *rip1Δ* mitochondria confirmed that Mar26 co-isolated mature as well as incompletely assembled complex III species containing cytochrome $c_1$ (Cyt1) (Fig. 4C). Similarly, Mar26 associated with not fully assembled complex III species in deletion mutants lacking the Rip1 biogenesis factors Bcs1 or Mzm1 (Wagener et al, 2011; Cui et al, 2012) (Figs. 4D and EV4).

Our findings strongly indicate that Mar26 may play a direct role in complex III biogenesis. To further explore this idea we imported radiolabeled preproteins of respiratory chain complex subunits into isolated wild-type and *mar26Δ* mutant mitochondria and followed their assembly into respiratory chain supercomplexes by blue native PAGE. Incorporation of Qcr10 that assembles at the final stages of complex III biogenesis with the 500 kDa intermediate (Smith et al, 2012; Ndi et al, 2018) was not affected in *mar26Δ* mitochondria (Fig. EV5A). Similarly, import and assembly of the complex IV subunits Cox13 into mature respiratory chain supercomplexes (Fig. EV5B), Cox5a into Cox1-containing assembly intermediates (Fig. EV5C), and of the late assembly factor Rcf1 (Fig. EV5D) was comparable in wild-type and *mar26Δ* mutant mitochondria. In marked contrast, incorporation of Qcr8 into respiratory chain supercomplexes was strongly impaired in the absence of Mar26 (Fig. 5A, lanes 4–6). Instead, radiolabeled Qcr8 accumulated in a complex of ~500 kDa. This assembly defect resembles the behavior of Qcr8 when imported into *rip1Δ* mitochondria (Fig. 5A, lanes 10–15), identifying the radiolabeled band as the 500 kDa late assembly intermediate of complex III. Surprisingly, radiolabeled, imported Rip1 assembled more efficiently into respiratory chain supercomplexes in the absence of Mar26 (Fig. 5B, lanes 4–6). Such increased labeling of respiratory chain supercomplexes is also observed when Rip1 is imported into *rip1Δ* mitochondria (Fig. 5B, lanes 10–12) (Wagener et al, 2011).

The accelerated assembly of Rip1 in *mar26Δ* and *rip1Δ* mitochondria suggests that the newly imported precursor takes over the role of the intrinsic protein that is missing (*rip1Δ*) or not sufficiently disposed (*mar26Δ*) for assembly into mature respiratory chain supercomplexes. Thus, Mar26 appears to maintain an assembly-competent pool of Rip1 and to regulate its association with the 500 kDa late complex III assembly intermediate.

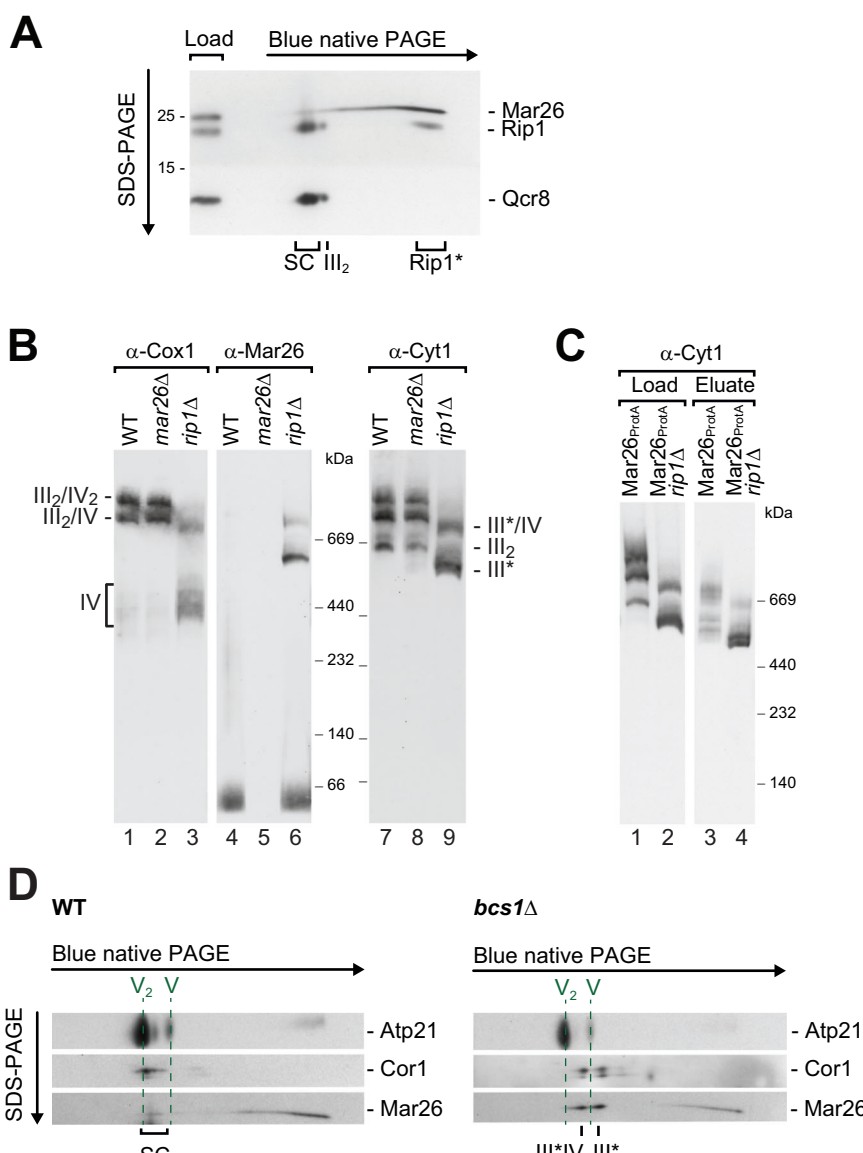

**Figure 4. Mar26 associates with Rip1 and partially assembled complex III.**

(A) Two-dimensional (native/non-native) gel analysis and western blotting showing the co-migration of Mar26 with Rip1 in the low molecular weight range of the native dimension. SC, supercomplexes, III$_2$, complex III dimer, Rip1*, Rip1 assembly intermediate. (B) Native protein complexes of wild-type, mar26Δ and rip1Δ mitochondria were analyzed by western blotting against the indicated proteins. III$_2$, complex III dimer; III*, 500 kDa complex III assembly intermediate; III*/IV, putative supercomplex of III* and IV. (C) Protein complexes of wild-type (WT) and Mar26$_{ProtA}$ were purified under native conditions and analyzed by SDS-PAGE or BN-PAGE and western blotting. Load, 1%; Eluate, 100%. (D) Wild-type or bcs1Δ mitochondria were solubilized and analyzed by two-dimensional (native/non-native) gel analysis as in (A). Monomeric (V) and dimeric (V$_2$) F$_1$F$_o$-ATP synthase complexes are indicated as internal markers. III$_2$, complex III dimer; III*, 500 kDa complex III assembly intermediate ; III*/IV, putative supercomplex of III* and IV; Atp21, F$_1$F$_o$-ATP synthase subunit; Cor1, complex III subunit. Source data are available online for this figure.

To narrow down at which stage of its biogenesis Rip1 interacts with Mar26, we imported radiolabeled Rip1 into bcs1Δ mitochondria in which its export from the matrix is blocked (Wagener et al, 2011), followed by a Mar26$_{ProtA}$ pulldown. Strikingly, lack of Bcs1 almost completely abolished the co-isolation of imported Rip1 with Mar26 (Fig. 5C). In the presence of Bcs1, the pulldown efficiency was significantly higher for newly imported Rip1 than for the endogenous pool, the majority of which is assembled into mature complex III and supercomplexes. These data suggest that Mar26 interacts with Rip1 selectively during its biogenesis, after Rip1 has

been inserted into the inner mitochondrial membrane by the AAA-ATPase Bcs1. Rip1 receives its Fe–S cluster in the mitochondrial matrix prior to Bcs1-dependent translocation of its redox domain into the intermembrane space (Wagener et al, 2011). To test whether Mar26 can interact with Rip1 lacking its Fe–S cluster, we imported radiolabeled Rip1 variants defective in Fe–S co-factor insertion (Rip1-S183C) (Denke et al, 1998) or lacking a structurally important disulfide bond (Rip1-C164Y/C180L) (Merbitz-Zahrad-nik et al, 2003), again followed by a Mar26$_{ProtA}$ pulldown. Both Rip1 mutants displayed strongly impaired binding to Mar26 upon

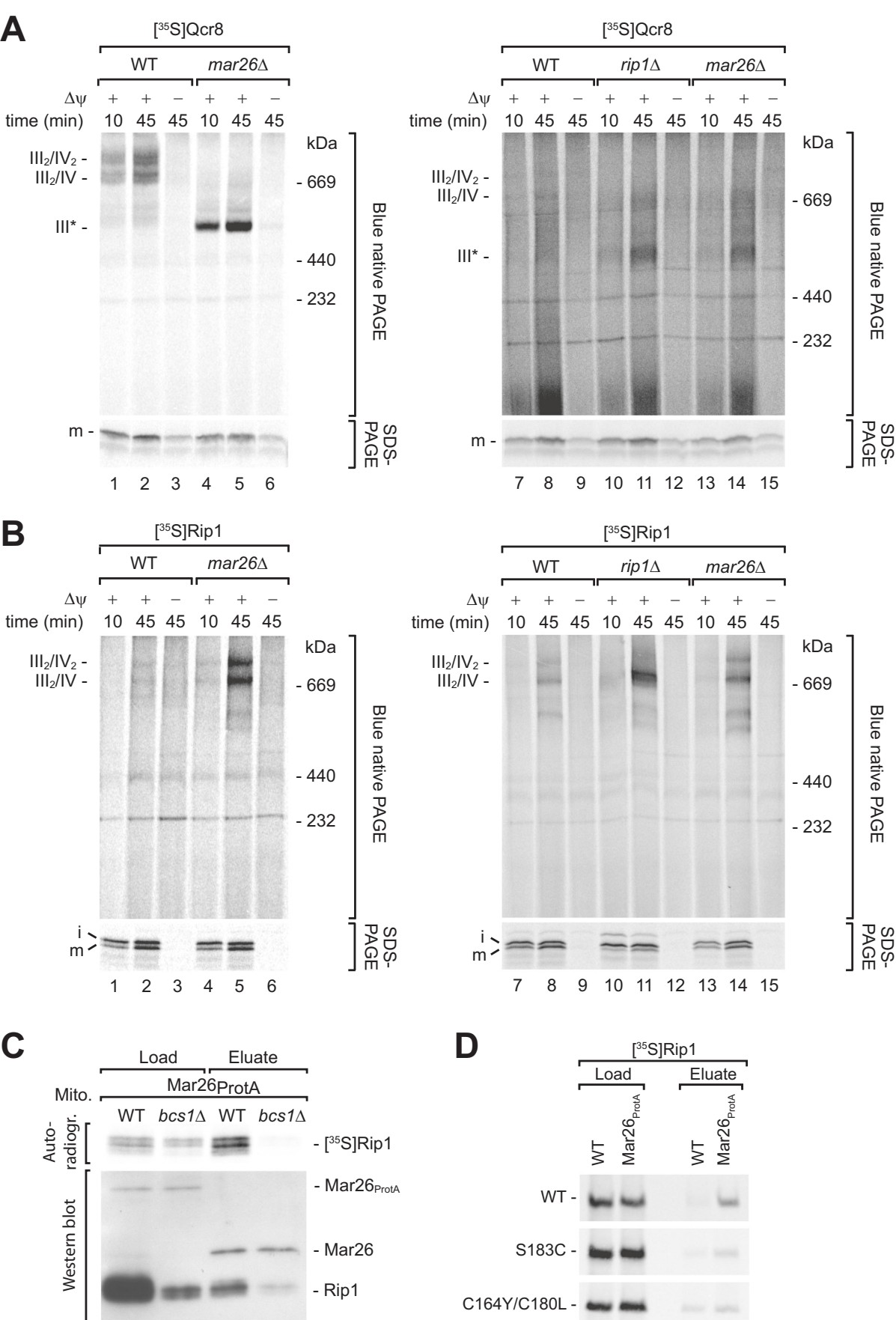

**Figure 5. Mar26 binds assembly-competent Rip1 to regulate complex III biogenesis.**

(A) Radiolabeled Qcr8 was imported into mitochondria isolated from wild-type (WT) or *mar26Δ* cells expressing Mic60$_{\text{ProtA}}$ grown on respiratory medium (lanes 1–6), or from WT, *mar26Δ* or *rip1Δ* cells grown on fermentative medium (lanes 7–15). Mitochondria were solubilized in digitonin-containing buffer and examined by SDS-PAGE or BN-PAGE and autoradiography. III$_2$/IV$_2$, III$_2$/IV, supercomplexes of respiratory chain complexes III and IV; III*, 500 kDa complex III assembly intermediate; Δψ, membrane potential; m, mature protein. (B) Radiolabeled Rip1 was imported into mitochondria and analyzed as in (A). i intermediate. (C) Radiolabeled Rip1 was imported into mitochondria isolated from wild-type (WT) or *bcs1Δ* cells expressing Mar26$_{\text{ProtA}}$. Mar26 and bound proteins were isolated by IgG chromatography, and the interaction of Mar26 with imported and endogenous Rip1 was assessed by autoradiography and western blotting. Load 3%, eluate 100%. (D) Radiolabeled Rip1 variants (Rip1 wild-type, Rip1-S183C, Rip1-C164Y/C180L) were imported into mitochondria isolated from wild-type (WT) or Mar26$_{\text{ProtA}}$ cells, Mar26 and interacting proteins were isolated by IgG chromatography, and the interaction of Mar26 with imported Rip1 variants was assessed by autoradiography. Load 8%; Eluate 100%. Source data are available online for this figure.

import into mitochondria (Fig. 5D), indicating that Mar26 selectively associates with correctly folded, co-factor-containing Rip1 at the inner mitochondrial membrane.

Taken together, we propose that the Mar26-dependent small Rip1 complex provides a constant pool of assembly-competent Rip1 for the late assembly stages of complex III biogenesis.

## Mar26 recruits imported Rip1 to MICOS for efficient assembly

We have demonstrated that Mar26 is a partner protein of MICOS and is also involved in late stages of complex III assembly by regulating the incorporation of Rip1. To ask whether these two findings reflect a functional connection, we examined the co-isolation of individual MICOS and respiratory chain subunits from Mic60$_{\text{ProtA}}$ *mar26Δ* mutant mitochondria in detail (Fig. 6A). We observed that among the respiratory chain subunits interacting with MICOS, solely Rip1 was co-isolated in reduced amounts when Mar26 was absent (Fig. 6A). The recovery in the elution fractions of all other tested MICOS and respiratory chain subunits was not changed for Mic60$_{\text{ProtA}}$ *mar26Δ* compared to Mic60$_{\text{ProtA}}$ mitochondria. Strikingly, overexpression of Mar26 in mitochondria increased not only the amount of Mar26 co-isolated with Mic60$_{\text{ProtA}}$ (Fig. 6A), but the amount of Rip1 associated with tagged Mic60 was also dramatically elevated (Fig. 6A). The co-purification of all other MICOS and respiratory chain subunits tested was not affected by increased Mar26 levels (Fig. 6A), and Mar26 overexpression did not cause any defects in respiratory growth (Fig. EV6A). Interestingly, radiolabeled Rip1 displayed impaired incorporation into mature respiratory chain super-complexes upon overexpression of Mar26 (Fig. EV6B). Together with the enhanced assembly of Rip1 in the absence of Mar26 (Figs. 5B and EV6B), this result reveals how Mar26 acts as a quality control factor that competes for interaction with newly imported Rip1. The physiological relevance of this quality control step is reflected by the reduced complex III activity and increased reactive oxygen species (ROS) production in *mar26Δ* mitochondria.

Since the Mar26-dependent small Rip1-containing complex interacts with MICOS (Fig. 1C, lane 29), we analyzed the elution fractions of MICOS complex isolations from *mar26Δ* or Mar26 overexpression mitochondria by BN-PAGE. In line with a Mar26-dependent recruitment of Rip1 to MICOS, our BN-PAGE analysis confirmed that increased levels of Mar26 resulted in a strong increase of the amounts of the small Rip1-containing intermediate complex in the elution fractions, demonstrating that unassembled Rip1 is more efficiently co-isolated with MICOS than mature respiratory chain complexes (Fig. 6B). We asked whether

recruitment of the Rip1 intermediate to MICOS is functionally significant. Loss of Mic60 does not result in accumulation of the late complex III assembly intermediate observed in *mar26Δ* mitochondria, nor is formation of the Mar26-dependent Rip1 intermediate affected in *mic60Δ* mitochondria (Fig. 3G), indicating that the Rip1-related function of Mar26 does not depend on MICOS. However, we found that assembly of newly imported Rip1 into mitochondria lacking Mic10 or Mic60 was impaired (Fig. 6C, lanes 1–9, Fig. 6D, left panel). Moreover, assembly of Qcr10, which is incorporated into complex III after Rip1, was impaired in MICOS-deficient mitochondria (Fig. 6C, lanes 10–18, Fig. 6D, right panel). In contrast, the early assembling subunit Qcr8 did not display a decreased assembly efficiency in MICOS mutant mitochondria (Fig. EV6C). We observed that the formation of the small Rip1 assembly intermediate is Mar26-dependent also in the case of newly imported Rip1 (Fig. EV6B, black circles). Notably, newly imported intermediate-bound Rip1 was co-isolated with MICOS much more efficiently than mature supercomplex-assembled Rip1, demonstrating that the co-isolation efficiency at steady state vastly underestimates the proportion of newly imported Rip1 bound to MICOS (Figs. EV6D and 6B, lane 5). Co-isolation of newly imported, intermediate-bound Rip1 with MICOS did not require completion of complex III biogenesis, as shown with *rho⁻* mitochondria (Fig. EV6D). We conclude that Mar26-mediated coupling of the Rip1 intermediate to an intact MICOS assures optimal spatial and temporal coordination of complex III assembly in mitochondria. Other assembly lines are possible under unfavorable conditions, but likely more error-prone, as suggested by the defect in complex III activity in the absence of Mic10 (Bohnert et al, 2015). This idea is supported by our finding that association of the Mar26/Rip1 intermediate with Mic60 is stablized in the absence of Mic10, indicating that the intermediate might be trapped at Mic60 in the absence of a fully functional MICOS (Fig. 6E).

Taken together, MICOS appears to act as an organizing platform that holds a reservoir of assembly-competent Rip1 and facilitates its assembly under conditions of ongoing complex III biogenesis. Mar26 interacts with a pool of unassembled Rip1 and connects it to the Mic60-Mic19 module of MICOS. In this way, Mar26 physically links late stages of complex III biogenesis to the MICOS complex at crista junctions.

## Discussion

Most respiratory chain complex subunits are nuclear-encoded and inserted into the inner membrane in the boundary region. The

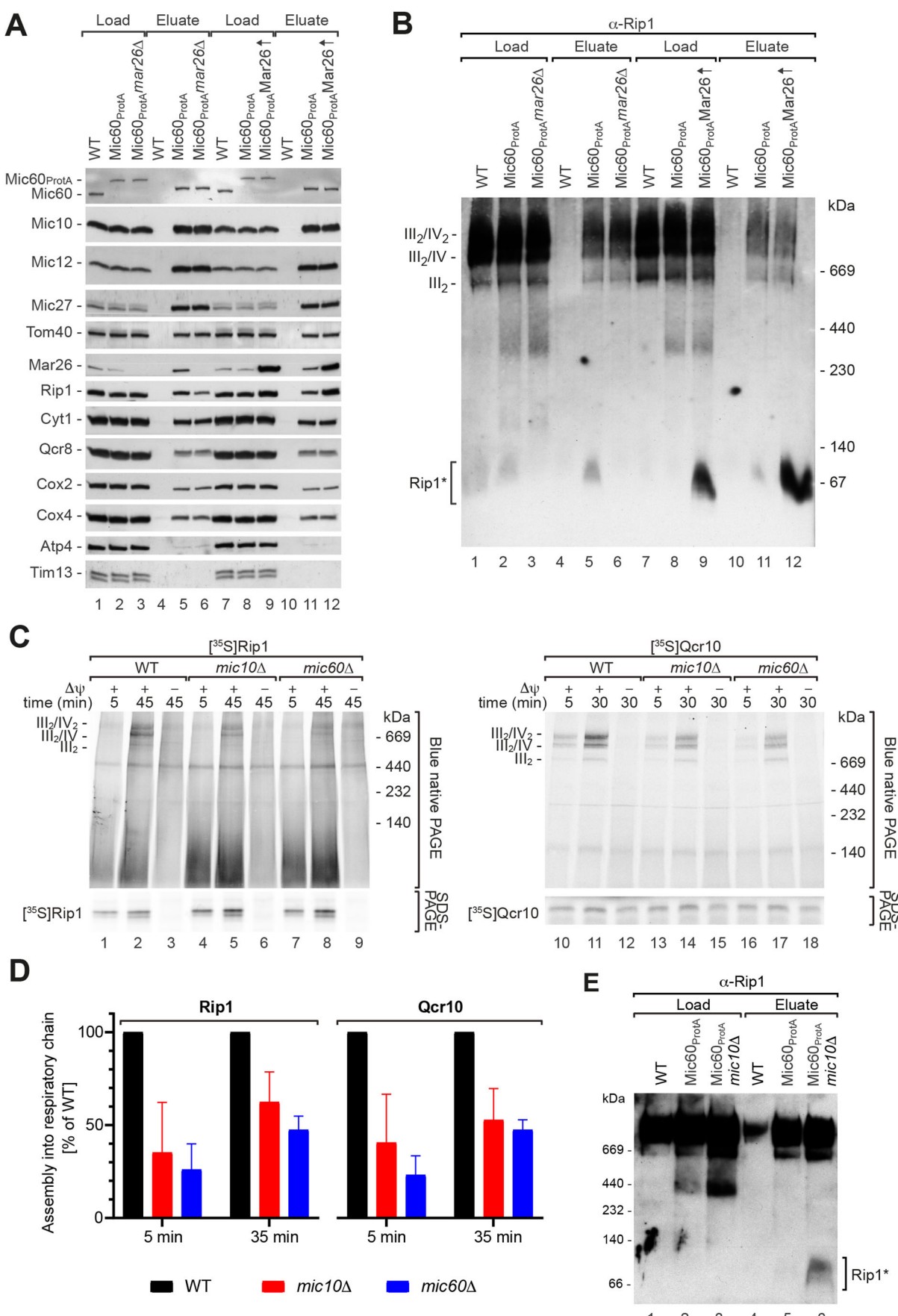

**Figure 6.   Mar26 recruits Rip1 to MICOS for facilitated assembly.**

(A) Protein complexes were purified from wild-type (WT), Mic60$_{ProtA}$, Mic60$_{ProtA}$ mar26Δ or Mic60$_{ProtA}$ Mar26 overexpression (↑) mitochondria by IgG chromatography. Samples were analyzed by SDS-PAGE and western blotting. Load, 4% (MICOS subunits, Tom40) or 1% (other proteins); Eluate, 100%. Atp4, F$_1$F$_o$-ATP synthase subunit. (B) Protein complex isolations were performed as in (A). Samples were analyzed by BN-PAGE and immunoblotting. III$_2$/IV$_2$, III$_2$/IV, III$_2$, supercomplexes formed by respiratory chain complexes III and IV; Rip1*, Rip1-containing assembly intermediate. Load, 1%; Eluate, 100%. (C) Radiolabeled Rip1 or Qcr10 were imported into isolated WT, mic10Δ or mic60Δ mitochondria. Mitochondria were solubilized in digitonin buffer, analyzed by SDS-PAGE or BN-PAGE and visualized by autoradiography. III$_2$/IV$_2$, III$_2$/IV, supercomplexes of respiratory chain complexes III and IV; Δψ, membrane potential. (D) Quantitation of Rip1 (left panel) or Qcr10 (right panel) assembly into respiratory chain (super)complexes (n = 3). Error bars: SD of the mean; n = 3 (3 independent experiments). (E) Protein complexes were purified from wild-type (WT), Mic60$_{ProtA}$ or Mic60$_{ProtA}$ mic10Δ mitochondria by IgG chromatography. Samples were analyzed by BN-PAGE and western blotting. Load 1%; Eluate 100%. Source data are available online for this figure.

assembly of these proteins with each other and with mitochondrially encoded subunits must be tightly coordinated in space and time (Richter-Dennerlein et al, 2016). Individual proteins or pre-assembled subcomplexes have to cross the permeability barrier at crista junctions to assemble into respiratory chain supercomplexes (Stoldt et al, 2018). Our results reveal that dedicated assembly factors bound to the MICOS complex at crista junctions play an important role in respiratory chain biogenesis, likely by facilitating submitochondrial protein sorting and coordination of complex assembly steps.

Recent work revealed a functional link between MICOS and respiratory chain complexes of mitochondria, but the molecular basis of this connection has remained unclear (Harner et al, 2014; Bohnert et al, 2015; Friedman et al, 2015; Guarani et al, 2015; Anand et al, 2020). Here we show that respiratory chain supercomplexes are specifically co-purified with MICOS complexes under mild detergent conditions, indicating a physical association of these two crucial mitochondrial protein machineries at the inner membrane. Defined assembly intermediates of respiratory chain complexes are present in mitochondria at low levels (Smith et al, 2012; Vercellino and Sazanov, 2022). Interestingly, some assembly intermediates such as the Mar26-dependent Rip1 intermediate discovered here appear to be enriched at MICOS complexes.

Our comprehensive MICOS and respiratory chain interactome analysis identified the so far uncharacterized protein Mar26 as a shared interaction partner. Deletion of MAR26 did not interfere with the coupling of respiratory chain supercomplexes to MICOS. Instead, we found that Mar26 plays a specific role in the biogenesis of complex III. We identified a small Rip1-containing subcomplex that likely represents an assembly intermediate and is only detectable in the presence of Mar26. The small Rip1 complex accumulates at MICOS and is connected to MICOS via Mar26. Because respiratory chain supercomplexes are still formed in vivo and only small amounts of partially assembled complex III accumulate in mar26Δ mutant mitochondria, Mar26 cannot be strictly required for complex III formation. Nevertheless, we show that loss of Mar26 limits respiratory growth, increases ROS production and considerably impairs complex III activity.

Our in vitro import and assembly assays shed light on a possible mechanistic role of Mar26 in complex III biogenesis. In these experiments, only individual subunits are imported into isolated mitochondria, whereas in vivo all subunits necessary for assembly of mature respiratory chain complexes are usually supplied together. The subunits must bind in a defined order to avoid disrupting the assembly process. Many dedicated assembly factors are known that orchestrate the assembly of respiratory chain complexes. We propose that Mar26 plays a specific role in the late

assembly steps of complex III. Our results suggest that Mar26 interacts with unassembled Rip1 and may constantly maintain a small, assembly-competent pool of this subunit. In mar26Δ mutant mitochondria, this pool of unassembled Rip1 is absent and a late complex III intermediate accumulates, leading to defective assembly of other subunits, like Qcr8, if supplied individually to the assembly line. In order to efficiently associate with Mar26, newly imported Rip1 must already contain its Fe–S co-factor that is added on the matrix side of the mitochondrial inner membrane. Moreover, it must have undergone the critical Bcs1-dependent export step exposing the redox center to the intermembrane space. Thus, our detailed analyses point to a defined step of Rip1 biogenesis that involves Mar26-dependent recruitment to MICOS. We propose that the Mar26-bound pool of Rip1 becomes critical under conditions where the balanced supply of subunits is disturbed—a condition that is mimicked in our in vitro assembly assay. In living cells, such unbalanced situations may occur stochastically with increased frequency when particularly high rates of respiratory chain complex biogenesis are required. It is therefore noteworthy that cells lacking Mar26 display a pronounced defect in the metabolic adaptation from fermentative to respiratory growth.

Our finding that respiratory chain intermediates interact with MICOS suggests that crista junctions may be a central hub for respiratory chain biogenesis. In fact, we demonstrate that beyond the role of Mar26 in Rip1 assembly, Mar26-dependent recruitment of the Rip1 intermediate to an intact MICOS is important for efficient Rip1 assembly. Significantly, MICOS preferentially interacts with newly imported, unassembled Rip1 compared to mature respiratory chain complexes. Upon import into MICOS-deficient mitochondria, incorporation of Rip1 into complex III is impaired. Delayed Rip1 assembly may result from loss of the interaction with the Mic60 subcomplex, or from the Rip1 assembly intermediate becoming trapped at a defunct MICOS complex in mic10Δ mitochondria. However, MICOS is not globally required for efficient complex III biogenesis since Qcr8 assembly is unaffected in MICOS mutants. Thus, the observed defects reflect a specific function of MICOS rather than a general disruption resulting from altered cristae architecture. In addition to direct actions of MICOS in recruiting assembly intermediates, the strong membrane curvature at crista junctions and the accumulation of specific phospholipids at these sites may also facilitate certain assembly steps (Horvath et al, 2015; Rampelt et al, 2017; Basu Ball et al, 2017; Anand et al, 2020).

We find that Mar26 is involved in the coordination of complex III assembly in space and time. Such a coordination requires not only chaperone-like proteins that facilitate the incorporation of

individual subunits, but also factors that prevent the premature assembly of subunits thereby acting as quality control factors. In line with this idea, newly imported Rip1 accumulates at Mar26, which is strongly enhanced by Mar26 overexpression leading to a delay in Rip1 assembly into mature complex III. Our data indicate that Mar26 recruits a fraction of unassembled Rip1 into a waiting position at MICOS, where specific assembly intermediates of respiratory chain complexes accumulate and may pick up further components. At this stage, we cannot rule out that Mar26 has additional roles in mitochondria beyond supporting Rip1 assembly. Like its paralog Mrx3, Mar26 belongs to a family of thioesterase domain-containing proteins with a mitochondrial localization (Brocker et al, 2010). A human member of this family, termed THEM5, has been linked to cardiolipin metabolism, but its specific function is unknown (Zhuravleva et al, 2012).

Based on our findings reported here, we propose that MICOS together with bound assembly factors acts as a coordinating hub in respiratory chain biogenesis. It is tempting to speculate that its function includes the sorting of mitochondrial inner membrane proteins, like respiratory chain complex subunits, between boundary and cristae membrane domains. Our study provides the first indications how mitochondrial ultrastructure and inner membrane protein biogenesis and distribution are linked to each other on the molecular level, and paves the way for further studies on the spatial and temporal organization of mitochondrial protein complex biogenesis.

# Methods

**Reagents and tools table**

| Reagent/resource | Reference or source | Identifier or Catalog number |
|---|---|---|
| **Experimental models** | | |
| WT (YPH499 – MATa; *ade2-101; his3-Δ200; leu2-Δ1; ura3-52; trp1-Δ63; lys2-801*) | Sikorski and Hieter, 1989 | 1501 |
| WT (BY4741 – MATa; *his3-Δ1; leu2-Δ0; met15Δ0; ura3-Δ0*) | Euroscarf | 1354 |
| Mic60$_{ProtA}$ (YPH499 *mic60::MIC60$_{ProtA}$-HIS3MX6*) | von der Malsburg et al, 2011 | 2035 |
| Mic12$_{ProtA}$ (YPH499 *mic12::MIC12$_{ProtA}$-HIS3MX6*) | von der Malsburg et al, 2011 | 3083 |
| Mic60$_{ProtA}$ *mic10Δ* (YPH499 *mic60::MIC60$_{ProtA}$-HIS3MX6; mic10::kanMX4*) | von der Malsburg et al, 2011 | 3049 |
| Mic12$_{ProtA}$ *mic10Δ* (YPH499 *mic12::MIC12$_{ProtA}$-HIS3MX6; mic10::kanMX4*) | This study | 4042 |
| Mic12$_{ProtA}$ *mic60Δ* (YPH499 *mic12::MIC12$_{ProtA}$-HIS3MX6; mic60::kanMX4*) | This study | 3093 |
| Mic60$_{ProtA}$ *mar26Δ* (YPH499 *mic60::MIC60$_{ProtA}$-HIS3MX6; mar26::kanMX4*) | This study | 3044 |
| Mar26$_{ProtA}$ (YPH499 *mar26::MAR26$_{ProtA}$-HIS3MX6*) | This study | 4047 |
| Mrx3$_{ProtA}$ (YPH499 *mrx3::MRX3$_{ProtA}$-HIS3MX6; arg4Δ*) | This study | 3579 |
| Cor1$_{TAP}$ (YPH499 *cor1::COR1$_{TAP}$-HIS3MX6*) | van der Laan et al, 2006 | 2510 |
| WT rho- (YPH499 rho-) | This study | 3777 |
| Mic60$_{ProtA}$ rho- (YPH499 *mic60::MIC60$_{ProtA}$-HIS3MX6; rho-*) | This study | 3778 |
| *mar26Δ* (YPH499 *mar26::kanMX4*) | This study | 3043 |
| *mrx3Δ* (YPH499 *mrx3::kanMX4*) | This study | 3575 |
| *mic60Δ* (YPH499 *mic60::kanMX4*) | von der Malsburg et al, 2011 | 3092 |
| *mic10Δ* (YPH499 *mic10::kanMX4*) | von der Malsburg et al, 2011 | 3047 |
| WT pRS426 (YPH499 pRS426) | Bohnert et al, 2015 | 3243 |
| *mar26Δ* pRS426 (YPH499 *mar26::kan MX4* pRS426) | This study | 5215 |
| Mic60$_{ProtA}$ pRS426 (YPH499 *mic60::MIC60$_{ProtA}$-HIS3MX6; pRS426*) | Bohnert et al, 2015 | 3244 |

| Reagent/resource | Reference or source | Identifier or Catalog number |
|---|---|---|
| Mic60$_{ProtA}$ Mar26 up (YPH499 mic60:: MIC60$_{ProtA}$-HIS3MX6; pRS426-MAR26) | This study | 3587 |
| Mar26$_{ProtA}$ rho- (YPH499 mar26::MAR26$_{ProtA}$-HIS3MX6; rho-) | This study | 3782 |
| Mrx3$_{ProtA}$ rho- (YPH499 ; mrx3::MRX3$_{ProtA}$-HIS3MX6; arg4Δ; rho-) | This study | 3783 |
| mar26Δ (BY4741 mar26::kanMX4) | Euroscarf | 2835 |
| mrx3Δ (BY4741 mrx3::kanMX4) | Euroscarf | 2833 |
| rip1Δ (BY4741 rip1::kanMX4) | Euroscarf | 1932 |
| bcs1Δ (BY4741 rip1::kanMX4) | Euroscarf | 5244 |
| Mic60$_{ProtA}$ rip1Δ (YPH499 mic60:: MIC60$_{ProtA}$-HIS3MX6; rip1::kanMX4) | This study | 4860 |
| Mar26$_{ProtA}$ rip1Δ (YPH499 mar26:: MAR26$_{ProtA}$-HIS3MX6 rip1::kanMX4) | This study | 4864 |
| Mar26$_{ProtA}$ bcs1Δ (YPH499 mic60:: MIC60$_{ProtA}$-HIS3MX6; bcs1::natMX6) | This study | 5838 |
| bcs1Δ (YPH499 bcs1::natMX6) | This study | 5837 |
| **Recombinant DNA** | | |
| pRS426-Mar26 | This study | RMZ-Mar26oe |
| **Oligonucleotides and show [QJ]other sequence-based reagents** | | |
| Rip1-SP6-F | GATCGATTTAGGTGACACTATAGAAGCGGCCACCATGTTAGGAATAAGATCATCTGTCAAGAC | This paper |
| Rip1-R | CTAACCAACAATGACCTTATCACCATC | This paper |
| Mar26-SP6-F | TCGATTTAGGTGACACTATAGAATACGCCGCCGCCATGTTTAAAAGAATCGCAATAGC | This paper |
| Mar26-R | CAGCTATTTTTCTTCTTTTGGATTCCC | This paper |
| Mrx3-SP6-F | TCGATTTAGGTGACACTATAGAATACGCCGCCGCCATGTCCAGAACTATTCCATTTC | This paper |
| Mrx3-R | CAGTCAAAACATATCAAGCCATTTAAA | This paper |
| Qcr8-SP6-F | GATCGATTTAGGTGACACTATAGAAGCGGCCACCATGGGTCCTCC | This paper |
| Qcr8-R | TTAAACATTAACTCTTTCCAGCTCTTCTC | This paper |
| Qcr10-SP6-F | TCGATTTAGGTGACACTATAGAAGCGGCCACCATGGCGTACACTTCT | This paper |
| Qcr10-R | TCAATTAGGTTTATCTTCTGG | This paper |
| Rcf1-SP6-F | TCGATTTAGGTGACACTATAGAATACGCCGCCGCCATGTCACGCATGCCATCTA | This paper |
| Rcf1-R | CAGTTACTTCTTTCCAAGCTTATTTTC | This paper |

| Reagent/resource | Reference or source | Identifier or Catalog number |
|---|---|---|
| Mar26-ProtA-F | AATGCCACGTTTTCTA GCGAACAAGGGAATCC AAAAGAAGAAAAACGTACGCT GCAGGTCGAC | This paper |
| Mar26-ProtA-R | GCTATTGTATAAATAAA TAAATAAATAAATAAGT AGACGGCAAATATCGA TGAATTCGAGCTCG | This paper |
| Mar26-del-F | TCTTGAGATGTT GTACTCTGAG | This paper |
| Mar26-del -F | TCTTATGGAAC AGGTGGCCT | This paper |
| bcs1D_S1_F | GCAGTAGACGTACCAA GGGACGATTTGAAAA AAAATGAAGTGTGCCCT CAATCTACCATGCGTA CGCTGCAGGTCGAC | This paper |
| bcs1D_S2_R | GCATCTTCTGAATTTTT ATTATATACAAGTTATG TAGTAAGGCGCCACTT AGCACCTAATCGATGA ATTCGAGCTCG | This paper |
| pGEM4Z_Rip1_fwd | AGGTGACACTATAGAA TACGATGTTAGGAATA AGATCATCTG | This paper |
| pGEM4Z_Rip1_rev | ATCCCCGGGTACCGAG CTCGCTAACCAACAAT GACCTTATC | This paper |
| Rip1_S183C_fwd | GTTCTGTCCTTGCCAT GGTTGCCATTAT GATATTTCCGGTAG | This paper |
| Rip1_S183C_rev | CTACCGGAAATATCAT AATGGCAACCATGGCA AGGACAGAAC | This paper |
| Rip1_C164Y_fwd | GGTATTTGTACTCACTT AGGTTATGTTCCAATTG GTGAAGC | This paper |
| Rip1_C164Y_rev | GCTTCACCAATTGGAA CATAACCTAAGT GAGTACAAATACC | This paper |
| Rip1_C180L_fwd | ATTTTGGTGGTTGGTT CTGTCCTTTACATGGTT CACATTATGATATTTCC | This paper |
| Rip1_C180L_rev | GGAAATATCATAATGTGA ACCATGTAAAGGAC AGAACCAACCACCAAAAT | This paper |
| **Chemicals, enzymes, and other reagents** | | |
| Digitonin | Matrix Biosciences | 60105 |
| [$^{35}$S]Methionine | PerkinElmer | NEG009T005MC |
| KOD Hot Start Master Mix | Novagen | 71842-3 |
| Flexi Rabbit Reticulocyte Lysate | Promega | L4540 |
| TNT Quick Coupled Reaction Mix | Promega | L2080 |
| CNBr-activated Sepharose 4B | GE Healthcare | 17-0430-01 |
| Mitochondrial CIII Activity Assay Kit | Sigma-Aldrich | MAK360 |
| Cytochrome *c* from equine heart | Sigma-Aldrich | C7752 |

| Reagent/resource | Reference or source | Identifier or Catalog number |
|---|---|---|
| Amplex Red $H_2O_2$/Peroxidase Kit | Invitrogen | A22188 |
| Dihydroethidium | Sigma-Aldrich | 37291 |
| **Software** | | |
| Adobe Illustrator | Adobe Inc. | n/a |
| Graphpad | Graphpad Software, LLC | n/a |
| Fiji | Schindelin et al, 2012 | n/a |

## Yeast growth and mitochondrial isolation

*Saccharomyces cerevisiae* strains (Reagents and Tools Table) used in this study are derivatives of either YPH499 (Sikorski and Hieter, 1989) (*MAT*a *ura3-52 lys2-801 ade2-101 trp1-Δ63 his3-Δ200 leu2-Δ1*) or BY4741 (*MAT*a *his3Δ0 leu2Δ0 met15Δ0 ura3Δ0*). Yeast strains were grown either in YPG medium (1% [w/v] yeast extract, 2% [w/v] bacto-peptone, 3% [v/v] glycerol), YPEG (YPG supplemented with 3% [v/v] ethanol), YPD medium (1% [w/v] yeast extract, 2% [w/v] bacto-peptone, 2% [w/v] glucose), YPGal medium (1% [w/v] yeast extract, 2% [w/v] bacto-peptone, 2% [w/v] galactose) or selective minimal medium (0.67% [w/v] yeast nitrogen base, 0.07% [w/v] CSM amino acid mixture minus uracil, 3% [v/v] glycerol) +/− 0.1% glucose at 30 °C, 34 °C or 37 °C. Growth conditions for heavy isotope labeling of yeast cells (SILAC) and for electron microscopy analysis are described separately. Growth tests were performed as serial dilutions spotted on agar plates, or in liquid media using a CLARIOstar plate reader (BMG Labtech) as described previously (Rampelt et al, 2022).

Crude mitochondria were isolated by differential centrifugation (Song and Rampelt, 2024). Briefly, cells were pre-treated with DTT buffer (100 mM Tris-$H_2SO_4$ pH 9.4, 10 mM DTT) and lysed with a glass Teflon homogenizer in homogenization buffer (100 mM Tris-HCl pH 7.4, 0.6 M sorbitol, 1 mM EDTA, 0.2% bovine serum albumine, 1 mM PMSF). Cell lysates were cleared by centrifugation steps at $2000 \times g$ and mitochondria were pelleted at $17,000 \times g$ and resuspended in SEM buffer (250 mM sucrose, 10 mM MOPS pH 7.2, 1 mM EDTA).

## Yeast strain construction

Strains expressing C-terminally Protein A-tagged versions of Mar26 or Mrx3 were generated by PCR-based homologous recombination. The cassette used encodes the protein A moiety, a Tobacco Etch Virus (TEV) protease cleavage site and a *HIS3* selection marker (Knop et al, 1999). Gene deletions were generated by amplifying the respective deletion cassettes from the corresponding Euroscarf strains or following Longtine et al, (1998). Cells that overexpress Mar26 were transformed with pRS426-Mar26 that contains the *MAR26* wild-type open reading frame (ORF) flanked by the endogenous promotor (500 bases upstream the ORF) and terminator (300 bases downstream the ORF).

All strains were constructed by lithium acetate transformation. To generate yeast cells lacking mtDNA (*rho*− strains), the respective strains were subjected to three passages on agar plates supplemented with ethidium bromide. Primers used are listed in the Reagents and Tools Table.

## Native analysis of protein complexes

For blue native PAGE, mitochondria were solubilized in solubilization buffer (1% [w/v] digitonin, 20 mM Tris-HCl, pH 7.4, 0.1 mM EDTA, 50 mM NaCl, 10% [v/v] glycerol, and 1 mM PMSF). After solubilization non-soluble debris was removed by a clarifying spin. Loading dye was added to the supernatant, and protein complexes were separated on blue native polyacrylamide gradient gels. For two-dimensional gel analysis, a BN-PAGE lane was excised and embedded horizontally in an SDS-PAGE gel. Custom-generated rabbit polyclonal antibodies were used, source: Rampelt and van der Laan labs.

## Affinity purification of protein complexes

For affinity purification of native mitochondrial membrane protein complexes mitochondria expressing Protein A-tagged versions of the indicated proteins and the corresponding wild-type mitochondria were solubilized in digitonin buffer (20 mM Tris-HCl pH 7.4, 50 mM NaCl, 0.1 mM EDTA, 10% [v/v] glycerol, 1% [w/v] digitonin, 2 mM PMSF, 1× Roche EDTA free protease inhibitor cocktail). After removing the debris by a clarifying spin ($12,000 \times g$, 10 min, 4 °C) detergent extracts were incubated with IgG-coupled Sepharose beads for 90-120 min. Extensive washing with washing buffer (20 mM Tris-HCl pH 7.4, 60 mM NaCl, 0.5 mM EDTA, 10% [v/v] glycerol, 0.3% [w/v] digitonin, 2 mM PMSF) removed unbound proteins from the column. Specifically bound proteins were eluted by TEV protease cleavage overnight at 4 °C in washing buffer. TEV elution fractions were either analyzed by SDS-PAGE or blue native PAGE.

## Protein import into isolated mitochondria

Radiolabeled precursors were generated by in vitro translation in the presence of [$^{35}$S]methionine (TNT SP6 Quick Coupled or Flexi Rabbit Reticulocyte Lysate Systems) and subsequently incubated together with mitochondria (50–100 µg per import reaction) diluted in import buffer (3% [w/v] bovine serum albumin, 250 mM sucrose, 80 mM KCl, 5 mM $MgCl_2$, 2 mM $KH_2PO_4$, 5 mM methionine, 10 mM MOPS–KOH pH 7.2, 2–4 mM ATP, 2–4 mM NADH, 5 mM creatine phosphate, 100 µg/ml creatine kinase) at 25 °C or 30 °C (Stojanovski et al, 2007; Priesnitz et al, 2020). Import reactions were stopped by the addition of AVO mix (8 µM antimycin A, 1 µM valinomycin, 20 µM oligomycin). Mitochondria were washed with SEM buffer and analyzed by SDS-PAGE or blue native PAGE followed by autoradiography.

## Protein localization

To assess the submitochondrial localization of proteins, mitochondria were suspended in SEM buffer (250 mM sucrose, 10 mM MOPS pH 7.2, 1 mM EDTA) and subsequently diluted 1:10 in EM buffer (10 mM MOPS pH 7.2, 1 mM EDTA). After 30 min incubation on ice, proteinase K was added to a final concentration of 25 μg/ml for another 15 min. Proteinase K was inactivated by the addition of 2 mM PMSF. To determine if proteinase K digested proteins upon lysis of mitochondria, mitochondria were solubilized in SEM supplemented with 0.5% [v/v] Triton X-100. Subsequently, proteinase K was added to a final concentration of 25 μg/ml. Proteinase K was inactivated by the addition of 2 mM PMSF. Proteins were analyzed by SDS-PAGE and western blotting. To separate soluble and membrane-associated proteins, isolated mitochondria were incubated in 0.1 M $Na_2CO_3$ at pH 10.8 or at pH 11.5 on ice for 30 min. Subsequently, membrane-bound proteins were pelleted by ultracentrifugation for 30 min at $100,000 \times g$ at 4 °C. Supernatant and pellet fractions were precipitated with trichloroacetic acid and analyzed by SDS-PAGE and western blotting.

## Enzymatic assays

Complex III activity was assayed with the mitochondrial complex III activity assay kit (Sigma-Aldrich) by measuring the initial reaction kinetics of cytochrome *c* (Sigma-Aldrich) reduction. Reactions were performed individually in 96-well plates and the absorption at 550 nm was recorded with a CLARIOstar plate reader (BMG Labtech). For Complex IV activity measurements, isolated mitochondria were lysed in assay buffer (10 mM Tris-HCl pH 7.4, 120 mM KCl) with 0.5% Triton X-100 and 8 μM antimycin A. They were then diluted in assay buffer with 7.5 μM (f.c.) reduced cytochrome *c* (Sigma-Aldrich), and the initial reaction kinetics of its oxidation were monitored at 550 nm in a PerkinElmer Lambda 35 Spectrometer. In gel complex IV activity was assayed by shaking the BN-PAGE gel in activity buffer (50 mM potassium phosphate pH 7.2, 1 mg/ml cytochrome *c*, 1.4 mM 3,3-diaminobenzidine tetrahydrochloride, 2 μg/ml catalase, 0.22 M sucrose) for 20 min at 30 °C. For in gel ATPase activity, the gel was incubated for 10 min in deionized water, then incubated for 45 min in ATP buffer (5 mM $MgCl_2$, 50 mM glycine, 20 mM ATP, pH 8.4). Detection was performed after rinsing and incubating the gel briefly in 10% [w/v] $CaCl_2$.

## Measurement of reactive oxygen species

In vivo $H_2O_2$ levels were detected using the Invitrogen Amplex™ Red Hydrogen Peroxide/Peroxidase Assay kit. Following the manufacturer's instructions, measurements over a time period of 120 min were performed at 30 °C in a 48-well plate using a TECAN Spark M1 plate reader. $H_2O_2$ levels were calculated using a $H_2O_2$ standard curve ranging from 0.05 to 1 μM $H_2O_2$. Superoxide formation was measured by dihydroethidium (Sigma-Aldrich) fluorescence. Isolated mitochondria were incubated for 10 min in the dark at 22 °C in MAS buffer (70 mM sucrose, 220 mM mannitol, 5 mM $KH_2PO_4$, 5 mM $MgCl_2$, 10 mM HEPES pH 7.2) supplemented with 5.5 mM succinate pH 7 and 2 μM (f.c.) dihydroethidium. Fluorescence was measured in a CLARIOstar plate reader (BMG Labtech) at an excitation of 480 nm and emission of 604 nm.

## Stable isotope labeling with amino acids in cell culture (SILAC)

Wild-type, Mic60$_{ProtA}$, Mar26$_{ProtA}$ and Cor1$_{TAP}$ yeast cells were grown in synthetic medium (0.67% [w/v] bacto-yeast nitrogen base, amino acid mix containing histidine, tryptophan, adenine, methionine, uracil, isoleucine, tyrosine, phenylalanine, leucine, valine, threonine and proline, 3% [v/v] glycerol) supplemented with 0.1 mg/ml ampicillin. Wild-type cells were labeled with 22.84 mg/l [$^{13}C_6/^{15}N_4$] L-arginine and 23.52 mg/l [$^{13}C_6/^{15}N_2$] L-lysine (Eurisotop, Gif-sur-Yvette, France) while Mic60$_{ProtA}$, Mar26$_{ProtA}$ and Cor1$_{TAP}$ cells were grown in media supplemented with the corresponding [$^{12}C/^{14}N$] amino acids (18 mg/l arginine and 22.5 mg/l lysine).

## LC/MS and MS data analysis

LC/MS sample preparation and analysis of affinity-purified MICOS complexes has been described before (von der Malsburg et al, 2011), and Cor1 and Mar26 complexes were processed essentially following the same protocol. In brief, differentially SILAC-labeled protein complexes were acetone-precipitated, proteins resuspended in 60% (v/v) methanol/20 mM $NH_4HCO_3$, and digested with trypsin. Peptide mixtures of three independent replicates per protein complex were analyzed on an LTQ-Orbitrap XL (Thermo Scientific, Bremen, Germany), which was directly coupled to an UltiMate™ 3000 HPLC or RSLCnano system (Thermo Scientific, Dreieich, Germany). Peptides were separated on C18 reversed-phase nano LC columns applying a 135- to 150-min linear gradient of increasing acetonitrile concentration [4 - 34% (v/v) in 0.1% (v/v) formic acid] at a flow rate of 300 nl/min (UltiMate™ 3000 HPLC) or 250 nl/min (UltiMate™ 3000 RSLCnano). MS survey scans ranging from *m/z* 300 to 2000 (Cor1) or 370 to 1700 (Mic60 and Mar26) were acquired in the orbitrap at a resolution of 60,000. Peptide ions with a charge of ≥ +2 were selected for fragmentation by collision-induced dissociation in the linear ion trap using a top6 method and applying a dynamic exclusion time of 45 s.

Mass spectrometric raw data of Cor1, Mic60, and Mar26 complexes were processed together using MaxQuant (version 1.4.1.2) and its search engine Andromeda (Cox and Mann, 2008; Cox et al, 2011). For protein identification, MS spectra were correlated with the *Saccharomyces* Genome Database (version of 02/03/2011). Heavy labels were set to Arg6/Lys6 for data derived from Cor1 complexes and Arg10/Lys8 for Mic60 and Mar26 data. The database search was performed with tryptic specificity; a maximum of two missed cleavages; acetylation of protein N-termini and oxidation of methionine as variable modifications; precursor and fragment ion mass tolerances of 4.5 ppm and 0.5 Da, respectively; a false discovery rate of 1% on peptide and protein level; and at least one unique peptide with a minimum length of 7 amino acids. Relative protein quantification was based on unique peptides and at least one SILAC peptide pair (i.e., ratio count). The data analysis is described in detail in the section Quantification and statistical analysis.

## Electron microscopy

For electron microscopy cells were inoculated in minimal medium (van Dijken et al, 1976) containing 2% (v/v) L-lactate (pH 5.0) and

0.1% glucose. After 24 h at 30 °C and 200 rpm cultures were diluted to an $OD_{600}$ of 0.1 in minimal medium containing only 2% (v/v) L-lactate (pH 5.0) as carbon source. After 16 h cells were harvested, washed with 0.1 M sodium cacodylate buffer pH 7.2 and subsequently fixed with 3% (v/v) glutaraldehyde in 0.1 M sodium cacodylate buffer (pH 7.2) for 1 h on ice. For diaminobenzidine (DAB) staining, fixed cells were incubated for 45 min in 0.1 M Tris buffer (pH 7.5) with 2 mg/ml 3:3'-diaminobenzidine and 0.06% (v/v) $H_2O_2$ at 30 °C under constant aeration. For post-fixation cells were incubated in 1.5% (w/v) $KMnO_4$ for 20 min at room temperature, followed by incubation in 0.5% (w/v) uranyl acetate overnight and embedding in Epon 812. For statistical analysis, 100 random cell sections were imaged and the number of crista junctions in each section was counted.

## Statistical analysis

For mass spectrometry analysis (Fig. 2A; Dataset EV1), MaxQuant results were further processed using an in-house developed data analysis pipeline programmed in R. In case no SILAC ratio was reported by MaxQuant, the missing value was estimated and replaced based on the distribution of proteins with the lowest 5% of MS intensity (Arg0/Lys0 label) of all proteins identified in a replicate. Following data imputation, ratios (light-over-heavy) were $log_2$-transformed, the mean $log_2$ ratios across all three replicates of a protein complex were calculated, and a one-sided $t$ test was performed to determine the $P$ value for each protein. Data about MaxQuant protein identification and quantification are provided in Dataset EV1. Data in Figs. 3B, D–F and EV3B include error bars depicting the SEM. Data describe biological and technical replicates. For the data in Fig. 3D,F, an unpaired two-tailed $t$ test was performed using the GraphPad software, each with a resulting $P < 0.001$. Quantitations shown in Fig. 6D include error bars depicting the SD of the mean.

## Data availability

This study includes no data deposited in external repositories.

The source data of this paper are collected in the following database record: biostudies:S-SCDT-10_1038-S44319-024-00336-x.

## Peer review information

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

## Acknowledgements

The authors thank Drs. Florian Wollweber, Susanne Horvath, Nikolaus Pfanner, and Nils Wiedemann for discussion, Thorina Boenke for materials, and Katja Noll and Sibylle Jungbluth for technical assistance. Work included in this study has also been performed in partial fulfillment of the requirements for the doctoral theses of RMZ, CSM, and CDP at the University of Freiburg. This project was supported by the Deutsche Forschungsgemeinschaft (DFG, German Research Foundation) FOR 2848—project ID 401510699 (to HR and MvdL), SFB 894—project ID 157660137 and SFB 746 - project ID 26068018 (to MvdL), the Excellence Initiative of the German Federal and State Governments (BIOSS—EXC-294 to MvdL and BW, and CIBSS— EXC-2189—project ID 390939984 to HR) and the European Research Council (ERC) Consolidator Grant 648235 (to BW).

## Author contributions

**Ralf M Zerbes**: Conceptualization; Resources; Formal analysis; Validation; Investigation; Visualization; Methodology; Writing—original draft. **Lilia Colina-Tenorio**: Resources; Formal analysis; Validation; Investigation; Visualization; Methodology. **Maria Bohnert**: Conceptualization; Resources; Formal analysis; Supervision; Validation; Investigation; Methodology. **Karina von der Malsburg**: Resources; Formal analysis; Supervision; Validation; Investigation; Visualization; Methodology. **Christian D Peikert**: Formal analysis; Validation; Investigation; Visualization; Methodology. **Carola S Mehnert**: Investigation; Methodology. **Inge Perschil**: Investigation; Methodology. **Rhena F U Klar**: Investigation; Methodology. **Rinse de Boer**: Validation; Investigation; Visualization; Methodology. **Anita Kram**: Validation; Investigation; Visualization; Methodology. **Ida van der Klei**: Supervision; Funding acquisition; Project administration. **Silke Oeljeklaus**: Formal analysis; Supervision; Validation; Investigation; Visualization; Methodology. **Bettina Warscheid**: Conceptualization; Formal analysis; Supervision; Funding acquisition; Validation; Project administration. **Heike Rampelt**: Conceptualization; Formal analysis; Supervision; Funding acquisition; Validation; Investigation; Visualization; Methodology; Writing—original draft; Project administration; Writing—review and editing. **Martin van der Laan**: Conceptualization; Formal analysis; Supervision; Funding acquisition; Validation; Investigation; Visualization; Methodology; Writing—original draft; Project administration; Writing—review and editing.

Source data underlying figure panels in this paper may have individual authorship assigned. Where available, figure panel/source data authorship is listed in the following database record: biostudies:S-SCDT-10_1038-S44319-024-00336-x.

## Funding

## Disclosure and competing interests statement

The authors declare no competing interests.

# Expanded View Figures

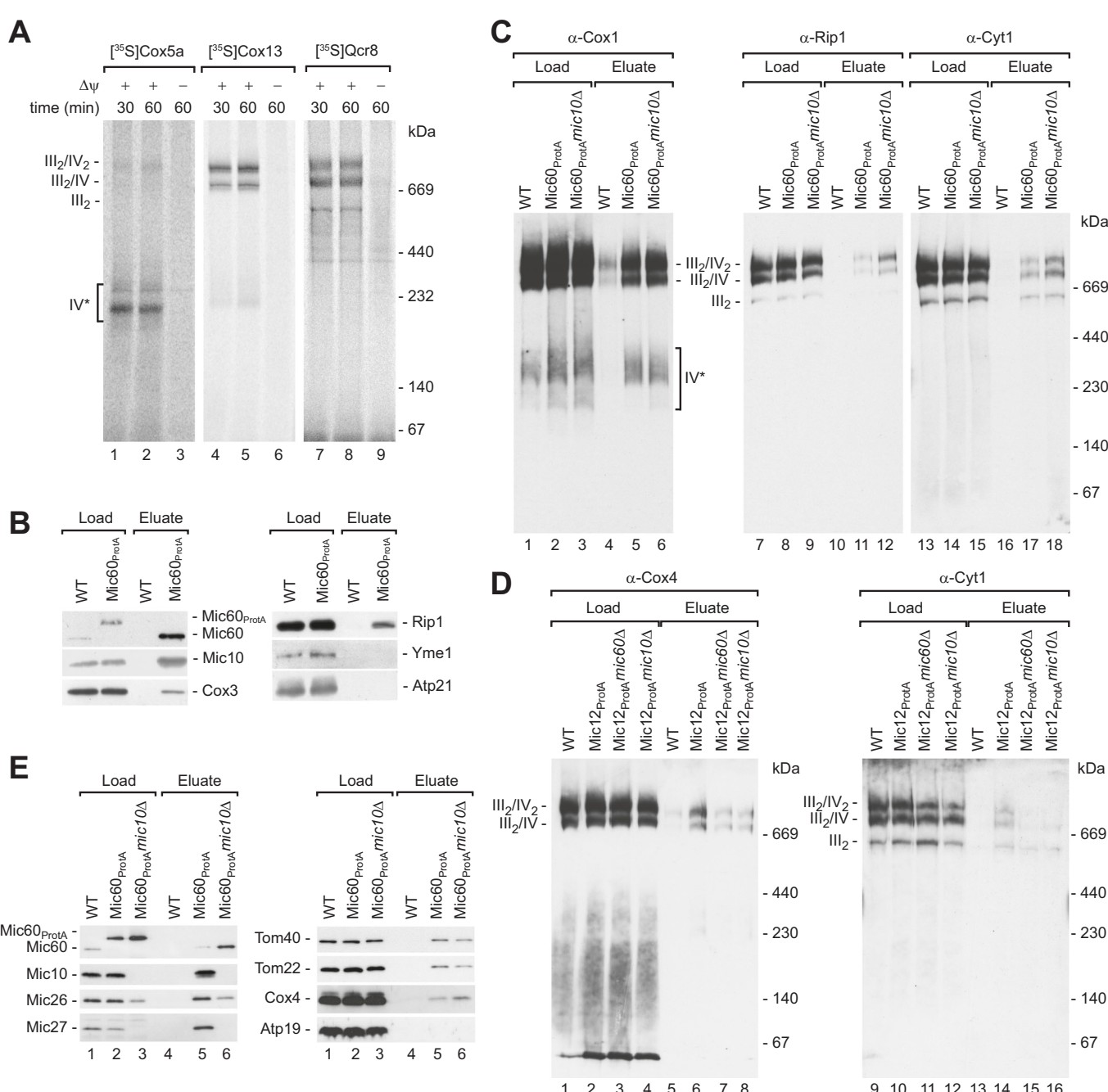

**Figure EV1. The Mic60-Mic19 module mediates MICOS connection to the respiratory chain.**

(A) The indicated radiolabeled preproteins were imported into wild-type mitochondria. Upon solubilization in digitonin-containing buffer, samples were analyzed by blue native (BN-) PAGE and autoradiography. $III_2/IV_2$, $III_2/IV$, $III_2$, supercomplexes of respiratory chain complexes III and IV; $IV^*$, complex IV and assembly intermediates thereof; $\Delta\psi$, membrane potential. (B) MICOS does not co-isolate the i-AAA protease Yme1 or the $F_1F_o$-ATP synthase subunit Atp21. IgG affinity purification was performed with wild-type (WT) and $Mic60_{ProtA}$ mitochondria and samples were analyzed by SDS-PAGE and immunoblotting. Load, 2%; Eluate, 100%. (C) Protein complexes were purified from digitonin-solubilized wild-type (WT), $Mic60_{ProtA}$ and $Mic60_{ProtA}$ mic10$\Delta$ mitochondria by IgG chromatography and analyzed by BN-PAGE and western blotting as in Fig. 1C. Load 1%, Eluate 100%. $III_2/IV_2$, $III_2/IV$, $III_2$, supercomplexes of respiratory chain complexes III and IV; $IV^*$, complex IV and assembly intermediates thereof. (D) Protein complexes purified from wild-type, $Mic12_{ProtA}$, $Mic12_{ProtA}$ mic60$\Delta$ and $Mic12_{ProtA}$ mic10$\Delta$ mitochondria were analyzed as in (A). Load 1%, Eluate 100%. Cox4, complex IV subunit. (E) IgG purification was performed with wild-type (WT), $Mic60_{ProtA}$ and $Mic60_{ProtA}$ mic10$\Delta$ mitochondria. Samples were examined by SDS-PAGE and immunoblotting. Load, 4.3%; Eluate, 100%. Source data are available online for this figure.

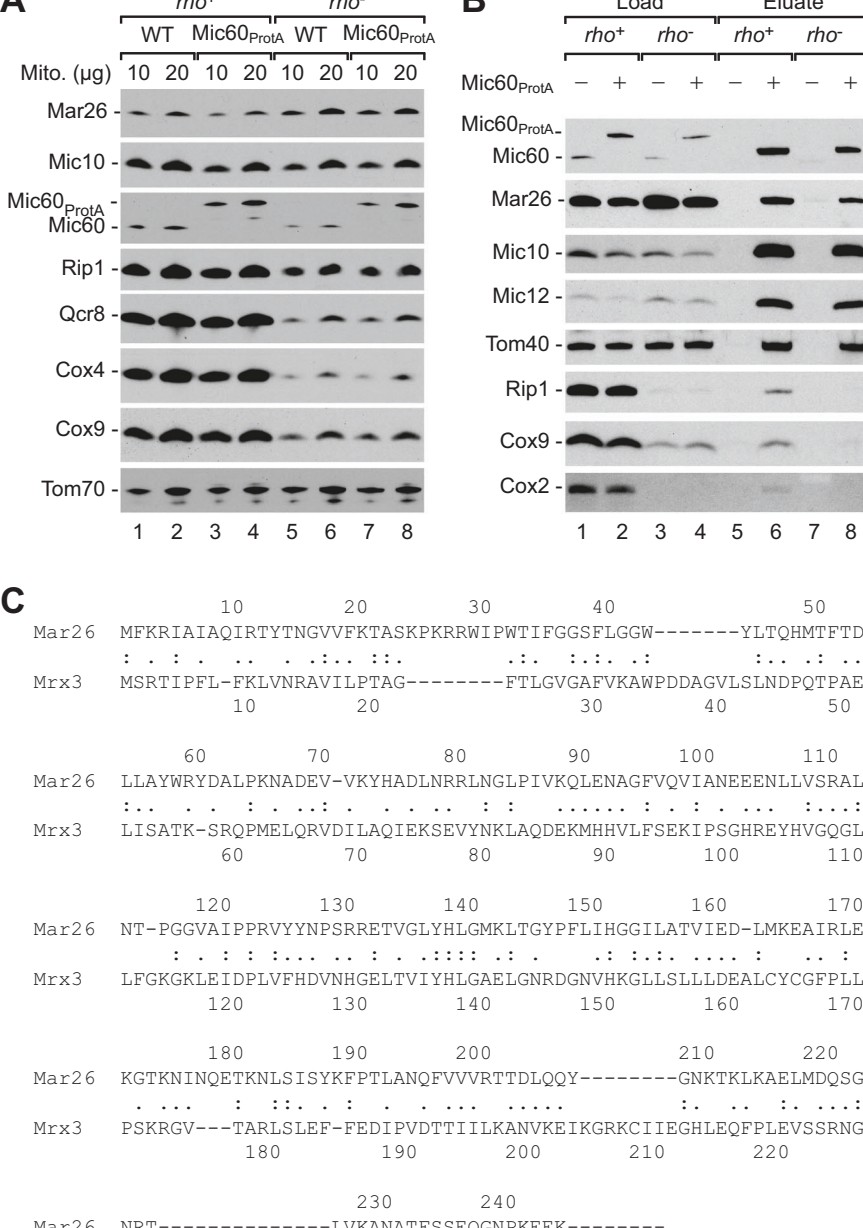

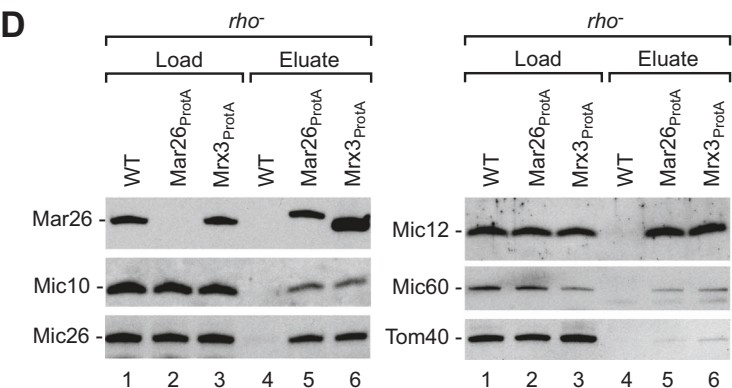

◀ **Figure EV2. Mar26 and Mrx3 are paralogs and interact with MICOS in the absence of a functional respiratory chain.**

(A) Steady-state protein levels of *rho*$^+$ or *rho*$^-$ mitochondria isolated from wild-type (WT) cells or cells expressing Mic60$_{ProtA}$ were analyzed by SDS-PAGE and western blotting with the indicated antibodies. (B) Protein complexes were purified from *rho*$^+$ or *rho*$^-$ WT or Mic60$_{ProtA}$ mitochondria by IgG chromatography and analyzed by SDS-PAGE and western blotting. Load, 1%; Eluate 100%. (C) Alignment of the amino acid sequences of Mar26 and Mrx3. (D) Protein complexes were purified from WT, Mar26$_{ProtA}$ and Mrx3$_{ProtA}$ mitochondria isolated from *rho*$^-$ cells by IgG chromatography and analyzed by SDS-PAGE and western blotting. Load, 1%; Eluate 100%.

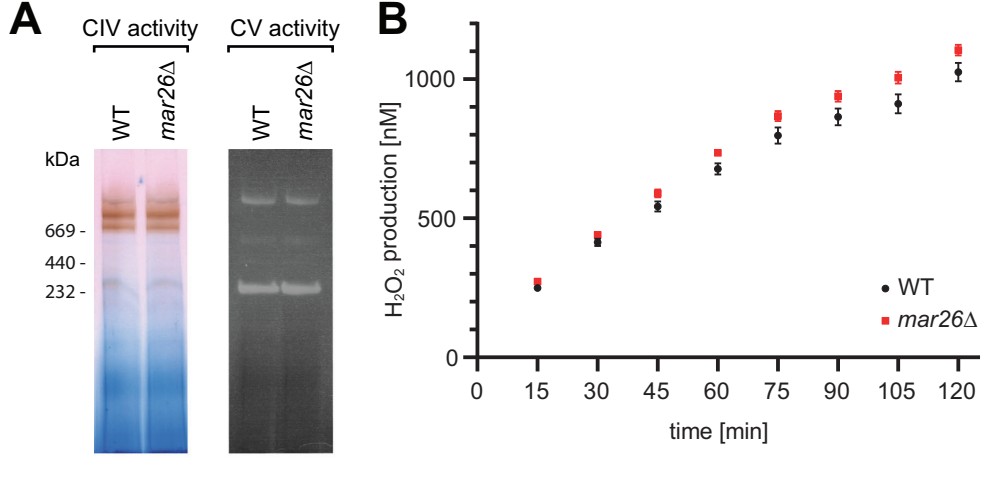

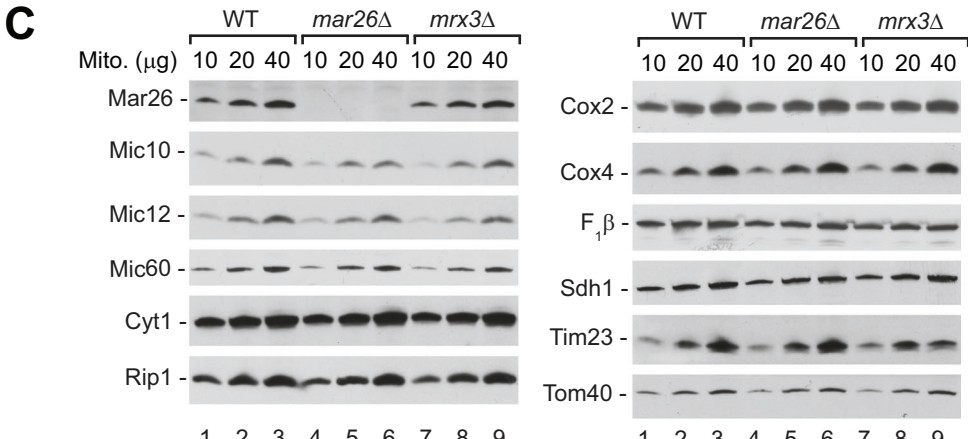

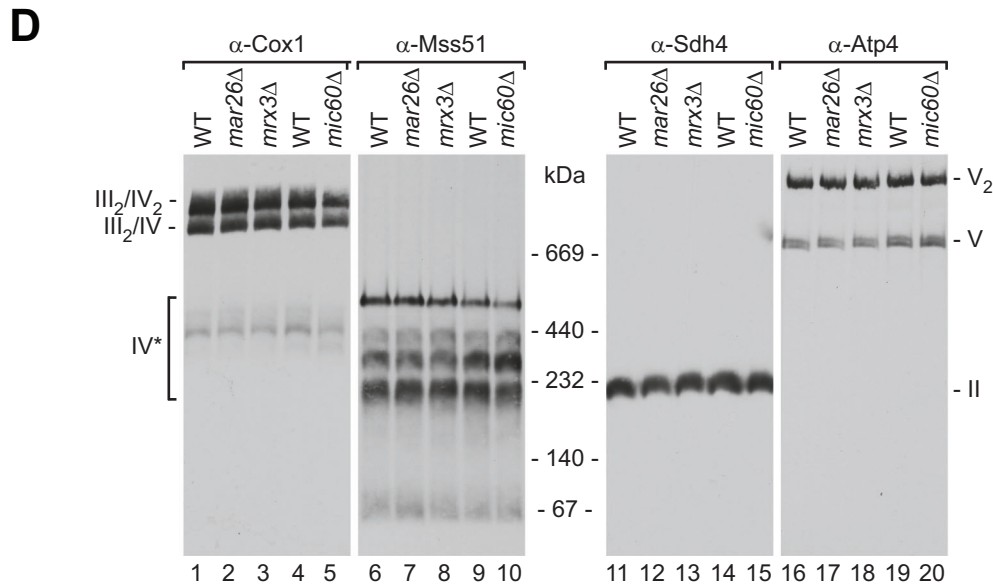

**Figure EV3. Mitochondrial protein or complex levels and activities in the absence of Mar26.**

(A) Wild-type or *mar26Δ* mitochondria were solubilized and protein complexes were separated by BN-PAGE, followed by the measurement of in gel activities for complex IV (left panel) or ATPase activity of the $F_1F_o$-ATP synthase complexes (complex V, right panel). (B) In vivo $H_2O_2$ levels in WT and *mar26Δ* cells were detected over a time period of 120 min at 30 °C using Amplex Red. $H_2O_2$ levels were calculated using a $H_2O_2$ standard curve. Error bars: SEM; $n = 5$ (technical replicates). (C) Steady-state protein levels of WT, *mar26Δ* and *mrx3Δ* mitochondria were analyzed by SDS-PAGE and western blotting. Sdh1, complex II subunit. (D) WT, *mar26Δ*, *mrx3Δ*, and *mic60Δ* mitochondria were solubilized in digitonin buffer. Protein complexes were subsequently separated by BN-PAGE and detected by western blotting and immunodecoration with the indicated antibodies. $III_2/IV_2$, $III_2/IV$, supercomplexes of respiratory chain complexes III and IV; IV*, complex IV and assembly intermediates thereof; II, respiratory chain complex II (SDH); $V_2$, V, $F_1F_o$-ATP synthase (complex V) dimers and monomers, respectively.

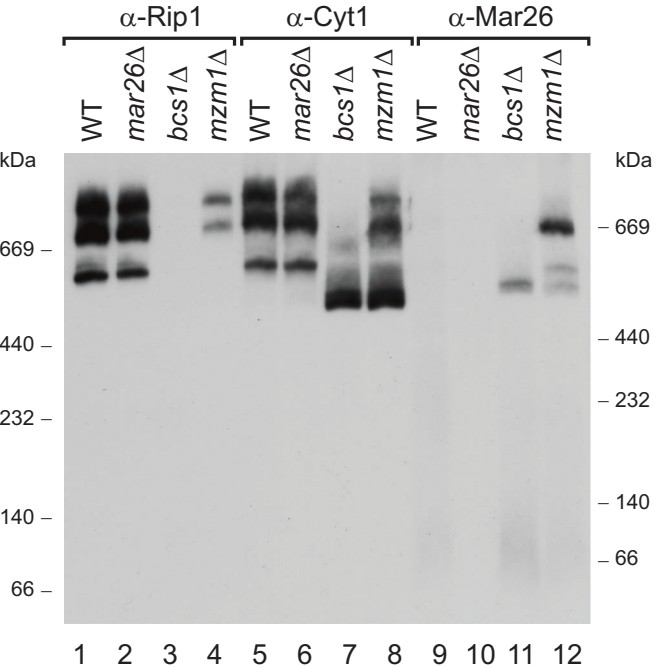

**Figure EV4. Steady-state levels of mitochondrial protein complexes in *mar26Δ* mutant mitochondria.**

Native protein complexes of wild-type, *mar26Δ*, *bcs1Δ* and *mzm1Δ* mitochondria were analyzed by western blotting against the indicated proteins.

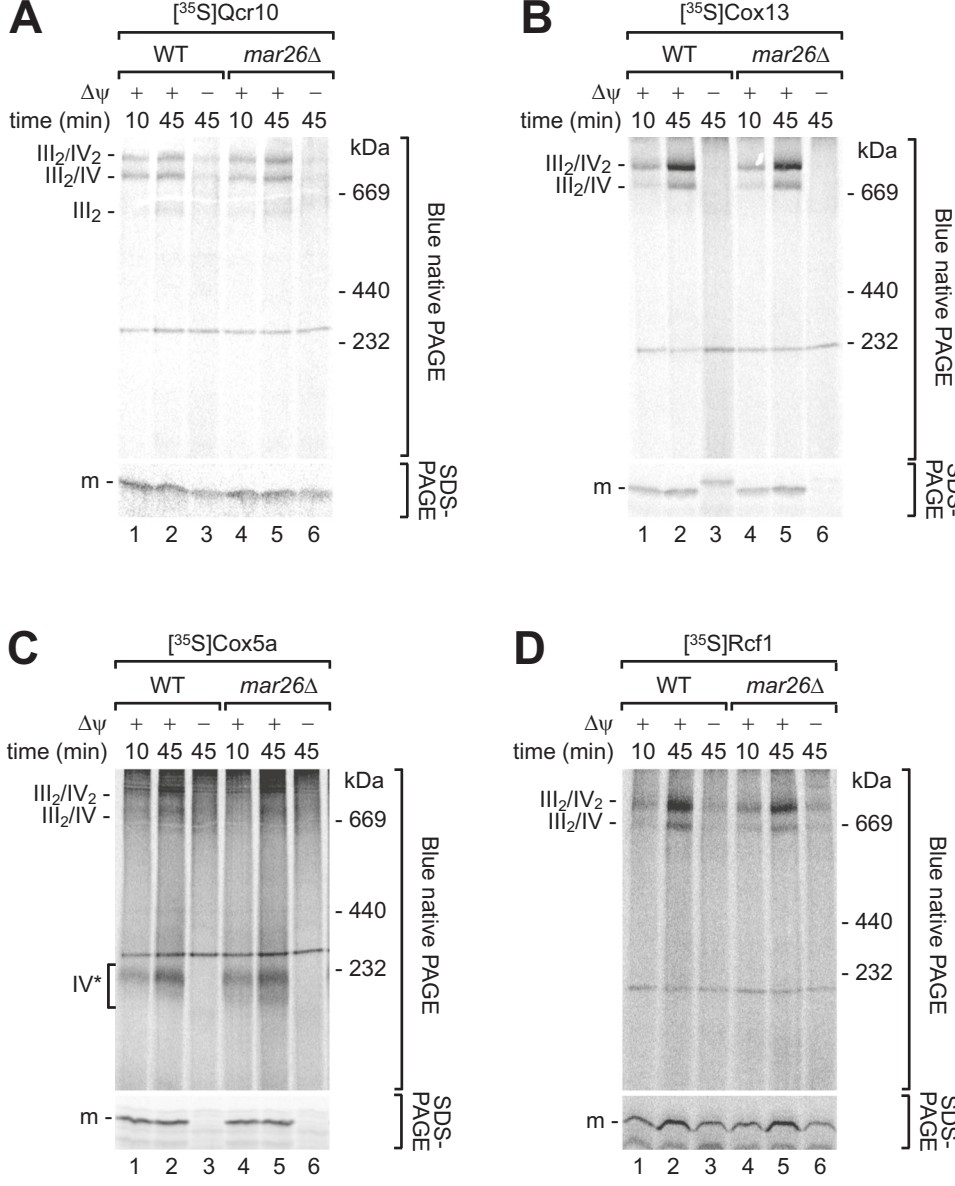

**Figure EV5. Assembly of respiratory chain complex subunits in *mar26Δ* mitochondria.**

Radiolabeled Qcr10 (**A**), Cox13 (**B**), Cox5a (**C**) or Rcf1 (**D**) preproteins were imported into wild-type or *mar26Δ* mitochondria in $Mic60_{ProtA}$ background. Samples were solubilized in digitonin buffer and analyzed by BN-PAGE or SDS-Page as indicated. Proteins were visualized by autoradiography. $III_2/IV_2$, $III_2/IV$, $III_2$, supercomplexes formed by respiratory chain complexes III and IV; IV*, assembly intermediates of complex IV; m, mature proteins; $\Delta\psi$, membrane potential.

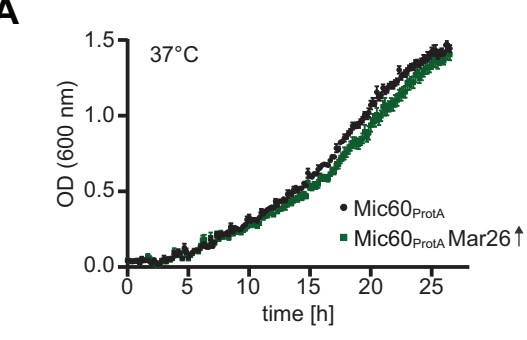

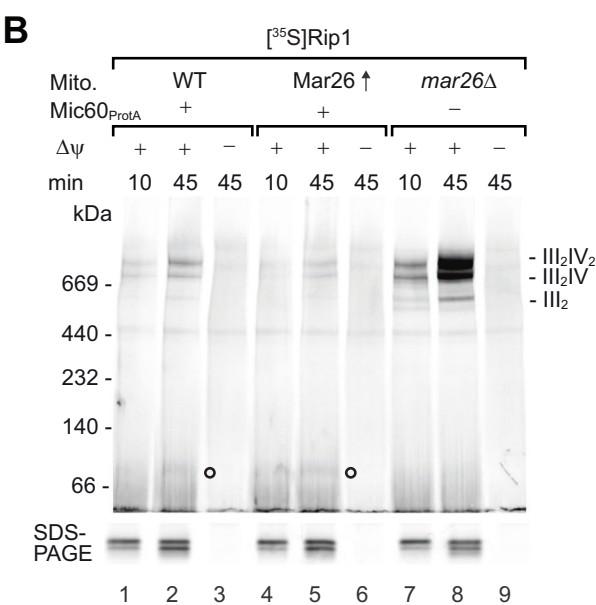

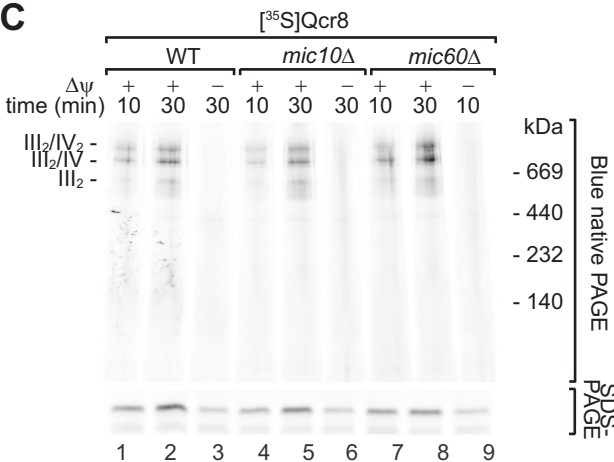

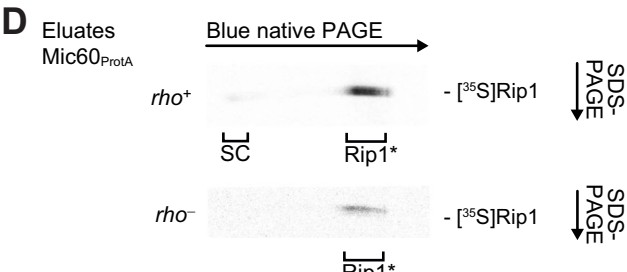

◀ **Figure EV6. Consequences of Mar26 overexpression, and MICOS binding the Rip1 intermediate.**

(A) Mic60$_{ProtA}$ and Mic60$_{ProtA}$ Mar26 overexpression (↑) cells were grown at 37 °C in liquid synthetic defined medium with 3% glycerol and 0.1% glucose as carbon sources. Error bars: SEM; $n = 8$ (2 independent experiments). (B) Radiolabeled Rip1 was imported into mitochondria isolated from Mic60$_{ProtA}$, Mic60$_{ProtA}$ Mar26 overexpression or $mar26\Delta$ cells. Mitochondria were subsequently solubilized in digitonin-containing buffer, analyzed by SDS-PAGE or BN-PAGE and visualized by autoradiography. Black circles, Rip1 intermediate; III$_2$/IV$_2$, III$_2$/IV, supercomplexes of respiratory chain complexes III and IV; $\Delta\psi$, membrane potential. (C) Radiolabeled Qcr8 was imported into mitochondria isolated from wild-type (WT), $mic10\Delta$ or $mic60\Delta$ cells. The samples were analyzed as in (B) and visualized by autoradiography. (D) Radiolabeled Rip1 preprotein was imported for 60 min into mitochondria isolated from Mic60$_{ProtA}$ mitochondria of $rho^+$ or $rho^-$ background, Mic60 and interacting protein complexes were isolated by IgG chromatography, and the eluates were subjected to two-dimensional gel electrophoresis. To monitor the interaction of MICOS with complexes containing newly imported Rip1, the membranes were assessed by autoradiography.

