## [Peer Review File · EMBO Reports]

Coordination of cytochrome bc1 complex assembly at MICOS

Ralf Zerbes, Lilia Colina-Tenorio, Maria Bohnert, Karina von der Malsburg, Christian Peikert, Carola Mehnert, Inge Perschil, Rhena Klar, Rinse de Boer, Anita Kram, Ida van der Klei, Silke Oeljeklaus, Bettina Warscheid, Heike Rampelt, and Martin van der laan

Corresponding author(s): Heike Rampelt (heike.rampelt@biochemie.uni-freiburg.de) , Martin van der laan (martin.van-der-laan@uks.eu)

Review Timeline:

Submission Date:	15th Apr 24
Editorial Decision:	23rd Apr 24
Revision Received:	7th Sep 24
Editorial Decision:	24th Oct 24
Revision Received:	4th Nov 24
Accepted:	15th Nov 24

Editor: Deniz Senyilmaz Tiebe

Transaction Report: This manuscript was transferred to EMBO reports following peer review at The EMBO Journal.

Dear Dr. Senyilmaz,

thank you for the constructive and very positive discussion this morning! Here is our summary of the revision plan as detailed earlier.

1) Role of Mar26 in CIII assembly / interaction with Rip1 / growth phenotype of mar26D (referee #1, point 3+6; referee #2, point 1+2, referee #3, point 2):

- complex III, IV, V activity measurements in mar26D mitochondria
- in vivo assessment of complex III deficiency via e.g. antimycin A sensitivity measurements
- Rip1 assembly in Mar26 overexpression mitochondria
- 2D gel analysis (BN-PAGE + SDS-PAGE) of the Rip1 intermediate in Mar26 overexpression mitochondria
- Rip1 import (wildtype and Fe-S cluster binding mutant) and pulldown of Mar26-ProtA; evtl. import into assembly-deficient mitochondria
- quantitative assessment of mar26D growth phenotypes in different respiratory media

2) Additional points:

- referee #1, point 2: controls for MICOS interactions
- referee #1, point 5: no renaming of YBL095W

3) Not essential but desirable: Role of MICOS in Mar26-mediated Rip1 assembly (referee #1, point 1; referee #2, point 3)

We already have data showing that the 500 kDa intermediate of complex III interacts with MICOS, and our preliminary data show that the Rip1 intermediate accumulates in mic10D mitochondria and is enriched in the Mic60-ProtA mic10D eluate compared to mature respiratory chain, supporting our conclusion that MICOS specifically facilitates Rip1 assembly from the Mar26-dependent intermediate.

We look forward to working with you for our revision!

Best wishes,

Heike

Dear Heike,
Dear Martin,

Thank you for transferring your manuscript to EMBO Reports along with referee reports from another venue, our video chat discussing your revision plan and sending your preliminary point-by-point response. I have now looked at your points carefully. I appreciate that you can address many of the concerns raised and see that the proposed experiments will strengthen the manuscript.

Having looked at everything, I would like to invite you to submit a revised manuscript. However, I would like to point out that we need strong support from the referees to consider publication here. As discussed during our video chat, the functional link between Mar26 and CIII assembly/function needs to be convincingly demonstrated as per referees. I am very glad to see that you are also willing to address the role of MICOS in Mar26 mediated in Rip1 assembly.

Please revise your manuscript with the understanding that the referee concerns (as in their reports) must be fully addressed and their suggestions taken on board. Please address all referee concerns in a complete point-by-point response. Acceptance of the manuscript will depend on a positive outcome of a second round of review. It is EMBO reports policy to allow a single round of major experimental revision only and acceptance or rejection of the manuscript will therefore depend on the completeness of your responses included in the next, final version of the manuscript.

We realize that it is difficult to revise to a specific deadline. In the interest of protecting the conceptual advance provided by the work, we recommend a revision within 3 months. Please discuss the revision progress ahead of this time with me if you require more time to complete the revisions, or if you have questions or comments regarding the revision (also by video chat).

1. A data availability section providing access to data deposited in public databases is missing (where applicable).
2. Your manuscript contains statistics and error bars based on $n=2$. Please use scatter plots in these cases.

You can submit the revision either as a Scientific Report or as a Research Article. For Scientific Reports, the revised manuscript can contain up to 5 main figures and 5 Expanded View figures, and it should not exceed 27000 characters. If the revision leads to a manuscript with more than 5 main figures it will be published as a Research Article. In this case the Results and Discussion section should be separate. If a Scientific Report is submitted, these sections have to be combined. This will help to shorten the manuscript text by eliminating some redundancy that is inevitable when discussing the same experiments twice. In either case, all materials and methods should be included in the main manuscript file.

4) a .docx formatted letter INCLUDING the reviewers' reports and your detailed point-by-point responses to their comments. As part of the EMBO publication's Transparent Editorial Process, EMBO reports publishes online a Review Process File (RPF) to accompany accepted manuscripts. This File will be published in conjunction with your paper and will include the referee reports,

your point-by-point response and all pertinent correspondence relating to the manuscript.

<https://www.embopress.org/page/journal/14693178/authorguide#transparentprocess>

5) a complete author checklist, which you can download from our author guidelines

<https://www.embopress.org/page/journal/14693178/authorguide>. Please insert information in the checklist that is also reflected in the manuscript. The completed author checklist will also be part of the RPF.

6) Please note that all corresponding authors are required to supply an ORCID ID for their name upon submission of a revised manuscript (<<https://orcid.org/>>). Please find instructions on how to link your ORCID ID to your account in our manuscript tracking system in our Author guidelines

<<https://www.embopress.org/page/journal/14693178/authorguide#authorshipguidelines>>

7) Before submitting your revision, primary datasets produced in this study need to be deposited in an appropriate public database (see <https://www.embopress.org/page/journal/14693178/authorguide#datadeposition>). Please remember to provide a reviewer password if the datasets are not yet public. The accession numbers and database should be listed in a formal "Data Availability" section placed after Materials & Method (see also

<https://www.embopress.org/page/journal/14693178/authorguide#datadeposition>). Please note that the Data Availability Section is restricted to new primary data that are part of this study. * Note - All links should resolve to a page where the data can be accessed. *

Additional information on source data and instruction on how to label the files are available:

<https://www.embopress.org/page/journal/14693178/authorguide#sourcedata>

9) Our journal encourages inclusion of *data citations in the reference list* to directly cite datasets that were re-used and obtained from public databases. Data citations in the article text are distinct from normal bibliographical citations and should directly link to the database records from which the data can be accessed. In the main text, data citations are formatted as follows: "Data ref: Smith et al, 2001" or "Data ref: NCBI Sequence Read Archive PRJNA342805, 2017". In the Reference list, data citations must be labeled with "[DATASET]". A data reference must provide the database name, accession number/identifiers and a resolvable link to the landing page from which the data can be accessed at the end of the reference. Further instructions are available at <http://www.embopress.org/page/journal/14693178/authorguide#referencesformat>

10) Regarding data quantification (see Figure Legends:

<https://www.embopress.org/page/journal/14693178/authorguide#figureformat>)

11) The journal requires a statement specifying whether or not authors have competing interests (defined as all potential or actual interests that could be perceived to influence the presentation or interpretation of an article). In case of competing interests, this must be specified in your disclosure statement. Further information: <https://www.embopress.org/competing->

interests

12) Please also note our reference format:

I look forward to seeing a revised version of your manuscript when it is ready. Please let me know if you have questions or comments regarding the revision.

Kind regards,

Deniz

Deniz Senyilmaz Tiebe, PhD
Scientific Editor
EMBO Reports

Referee #1:

In this work, the authors suggest a novel role of the MICOS complex in respiratory chain biogenesis. Through biochemical approaches, they demonstrate that MICOS and respiratory complexes are physically associated. Using SILAC of Cor1, they identify an uncharacterized mitochondrial protein (Fmp10/Mar26) as a new factor involved in Complex III biogenesis, which they suggest regulates an assembly intermediate with Rip1. Mar26 had been previously seen in SILAC of MICOS subunits, and the author explore whether Mar26 works with MICOS to promote Complex III assembly at cristae junctions. They show data that loss of MICOS subunits Mic10 and Mic60 causes a slight reduction in assembly of multiple Complex III proteins.

This is an interesting study that reports several novel findings including the identification of a novel determinant, Mar26, in Complex III biogenesis and a physical link between the MICOS and respiratory complexes. The experiments are well-performed and the data are of high quality. However, the functional requirement of MICOS in Rip1-Mar26 mediated complex III biogenesis is not supported by the data shown. In addition, while loss of Mar26 causes a respiratory growth defect, the effects on Complex III assembly are modest and do not necessarily explain a direct functional role of the protein in Complex III biogenesis. At this point, the data are too preliminary to support renaming Fmp10 as Mar26.

Specific points:

1. While the authors convincingly demonstrate that Mic60 can interact with Complex III subunits, the model that this interaction locally mediates Complex III assembly at cristae junctions as a means to pass the diffusion barrier is not supported. The only functional role of MICOS shown by the authors is an extremely subtle and non-quantitative Complex III assembly defect. This defect is found in the absence of both Mic60 and Mic10, which implies it is due to a general consequence of MICOS disruption rather than a specific role of the Mic60-Mar26 interaction. Besides a physical interaction between Mic60 and subunits of Complex III, there are no orthogonal assays that argue that Mar26 promotes Complex III biogenesis in concert with, or in proximity to, MICOS.
2. A concern is that the MICOS interaction with respiratory complexes, while convincingly shown, may not be specific or physiologically relevant. Controls with other abundant boundary membrane proteins such as Tim complexes and Yme1 should be shown.
3. While Mar26 expression levels correlate with the presence and absence of a Rip1 assembly intermediate, and loss of Mar26 leads to the subtle accumulation of a Complex III* intermediate, there is no evidence that these minor changes explain the respiratory growth defect of mar26-delta cells. As a whole, Complex III assembly looks relatively unaltered in the absence of Mar26. Is Complex III activity specifically altered in the deletion and could this explain the respiratory defect?
4. Mar26 may also indirectly cause the defects in Complex III assembly. While SILAC analysis of Mar26 identifies respiratory complex components, these interactions may not be specific. The authors show accumulation of a small Mar26 complex by blue native PAGE that they claim is the size of Rip1*. Is this dependent on Rip1 or Complex III assembly?
5. While the authors provide evidence that YBL095W shares minimal homology with Mar26, this does not justify renaming the protein as Mar30. Additionally, it is not clear whether YBL095W contains a transmembrane domain, which may explain the solubility of the protein relative to Mar26 in alkaline extraction assays. Thus, YBL095W may be an intermembrane space localized protein with a completely different functional role.

6. Can the overexpression of Mar26 be used to provide more functional information on the protein or be used to test their model. For example, does the accumulation of a Rip1* intermediate allow detection of a Rip1*-Mar26 complex? Does it titrate Rip1 from the assembled Complex III or lead to accumulation of Complex III*?

Minor Points

1. Why do Mic60-ProtA bands in the elution fraction run at a different sizes as compared to Mic60-ProtA in the load fraction? Is the protein A tag cleaved off during the elution?
2. Mic12 -ProtA western blots should be shown in Figure 1A.
3. Typographical error on Page 13, Line13 - "und" should be "and".
4. Typographical error on the top of page 15 - reference to Fig. S4C should be S4B

Referee #2:

In this work, Zerbes et al characterize the mitochondrial protein Fmp10 (which they now refer to Mar26) and suggest that it plays an important role in late stages in the biogenesis of respiratory complex III. The protein also co-purifies with the MICOS complex, which is located at the junction between the inner boundary membrane and the cristae, invaginations of the inner membrane. Based on this, the authors suggest that late stages complex III biogenesis at this site could be a possibility to determine the distribution of respiratory chain complexes between the different compartments of the inner membrane. Overall, the technical quality of the experiments is high and the manuscript is well written. However, a few key experiments are missing, limiting the molecular insights into the function of Mar26 and questioning the overall significance of the study. Hence, the authors need to experimentally address a number of key questions, the results of which could potentially require a substantial revision of the text.

Major points:

1. The exact molecular function of Mar26 is yet poorly defined. While absence of Mar26 provokes a massive growth defect on respiratory medium (Fig 5A), the accumulation of respiratory chain enzymes including complex III is not affected (Fig 5C and D). This would suggest that Mar26 does not play an important role in complex III biogenesis, but rather in something else. In order to substantiate that absence of Mar26 provokes a specific defect in complex III biogenesis, the authors should provide data showing the specific enzymes activities of oxidative phosphorylation complexes from WT and mar26D mitochondria. Specifically, the authors should determine quinol-dependent cytochrome c reductase activity for complex III, KCN-sensitive cytochrome c oxidation for complex IV and oligomycin-sensitive ATPase activity. If these experiments reveal that complex III activity is unchanged or not substantially reduced, the conclusion could be that the slight changes in Rip1 biogenesis in absence of Mar26 could be a secondary consequence of Mar26 loss, requiring then the identification of the primary function of Mar26 in mitochondria.
2. The authors suggest that Mar26 plays a role in Rip1 incorporation into complex III. Their most significant result is that a small complex of Rip1 is absent in BN-PAGE in the mutant lacking Mar26 (Fig 5D). The composition of this minor Rip1 complex is yet unknown, but it is possible that Mar26 is one of the components. Hence it would be important to isolate Mar26 complexes and determine their composition by MS and through antibodies. Moreover, there is a whole literature on Rip1 biogenesis, where the protein is first completely imported into the matrix, where it receives the Fe-S cluster, which is stabilized by Mzm1. Next, the mature Rip1 is translocated by the ATPase Bcs1, which associates with the 500 kDa intermediate to facilitate Rip1 assembly into this complex. Data from this study shows that incorporation of newly imported Rip1 into the complex is accelerated upon deletion of Mar26 (Fig 6B), suggesting that presence of Mar26 might be inhibiting assembly of newly imported Rip1. Hence, the exact role of Mar26 is yet ill-defined. The authors should test whether Mar26 directly and specifically interacts with newly imported Rip1 and test whether Rip1 would require the residues necessary for coordination of the Fe-S cluster for this interaction. Likewise, it would be important to elucidate the relationship of Mar26 with the known Rip1 biogenesis factors Mzm1 and Bcs1. Do they collaborate in Rip1 biogenesis, or do they work on alternative pathways?
3. The authors suggest that the MICOS complex, through interaction with assembly factors like Mar26, facilitates respiratory complex assembly specifically to cristae junctions, likely to distribute fully assembled complexes into cristae membranes. While this is an appealing hypothesis, this work does not provide substantial evidence that this is the case. If MICOS would play a prominent role in organizing late-stage complex III assembly, one would expect that MICOS' interactions with assembly factors and assembly intermediates would be rather quantitative. However, data presented in Fig 1 and 2 show that MICOS primarily interacts with fully assembled complex III (found in supercomplexes) and not with the 500 kDa intermediate of complex III assembly. Likewise, it is not known how much of Mar26 and the small Rip1 intermediate is bound to MICOS. Data presented in Fig S4C shows that Mar26 forms several separate complexes. The authors should determine the migration pattern of Mar26 and MICOS complexes from comparative co-migration analyses using BN-PAGE and test how Mar26 complexes change in complex III assembly or MICOS mutants.

Referee #3:

The study by Zerbes et al. addresses the interplay between cristae organization and assembly of respiratory (super-)complexes. The authors show that the MICOS complex is in physical contact to respiratory supercomplexes and they identify a novel player, Mar26 (and also its homolog Mar30), which specifically helps to maintain an assembly-competent Rip1 pool which appears to promote supercomplex assembly in the local neighbourhood of the MICOS complex at crista junctions. Moreover, the authors show that deletion of MICOS subunits alters the assembly kinetics of distinct subunits of supercomplexes in a specific manner. Overall, I think that this is an excellent and very important study which provides fundamental insights into how complex assembly is mechanistically linked to cristae organization. There is indeed really very little known in this regard. It reveals a specific novel pathway for a late-assembly step in complex III assembly linked to the MICOS complex. Moreover, the data shown is very convincing and very well described. I really only have minor suggestions that might improve the study further.

1. The assembly kinetics of Rip1 into complex was proposed to be impaired in strains lacking Mic10 or Mic60. This is very interesting and could indeed be true. Yet, it is not fully convincing and should be quantified (see Fig. 7C).
2. Fig. 1C. (lanes 23/24/29/30). The Rip1-containing complex at 140 kDa in the elution fraction is difficult to see, even in the long exposure. This could be improved (compare Fig. 7B lane 2). In addition, I suggest to decorate also the blot of the BN-PAGE (Fig. 7B with mar26-delta and mar26-OE) with the Mar26-antibody testing whether Mar26 also has a native size of around 140 kDa. This would strengthen the interpretation that Mar26 and Rip1 are forming an assembly-competent intermediate.
3. Fig. 2A. Related to this also a long exposure of Rip1 in Fig. 2A would be helpful to see whether deletion of Mic10 has an effect on the 140 kDa intermediate.
4. Fig. 4B. Here Cyt1 is shown. What about the effect of loss of Mar26 on the interaction of Mic60 with Rip1 or Cox1?
5. The homologs of Mar26 in animals are mitochondrial thioesterases. Do the authors think that this activity and its link to fatty acid metabolism is relevant. I suggest to discuss this aspect in the discussion section.

Response to referees

We would like to thank the referees and the editor for their constructive evaluation of our manuscript. We have followed all their suggestions and, as a consequence, have revised our study thoroughly with a substantial number of new experiments shown in Figs. 3B, 3D-F, 4A-D, 5C-D, 6D-E, as well as in Figs. EV1B, EV3A-B, EV4, EV6B+D. In our view the new findings strongly support our original conclusions. The revised manuscript establishes more clearly and convincingly both the role of Mar26 in complex III biogenesis and the contribution of MICOS to assembly efficiency. We hope that our revised manuscript now meets the high standards for publication in *EMBO Reports*. Please find below our point-by-point response to the referees' comments.

Referee #1:

In this work, the authors suggest a novel role of the MICOS complex in respiratory chain biogenesis. Through biochemical approaches, they demonstrate that MICOS and respiratory complexes are physically associated. Using SILAC of Cor1, they identify an uncharacterized mitochondrial protein (Fmp10/Mar26) as a new factor involved in Complex III biogenesis, which they suggest regulates an assembly intermediate with Rip1. Mar26 had been previously seen in SILAC of MICOS subunits, and the author explore whether Mar26 works with MICOS to promote Complex III assembly at cristae junctions. They show data that loss of MICOS subunits Mic10 and Mic60 causes a slight reduction in assembly of multiple
Complex III proteins.

This is an interesting study that reports several novel findings including the identification of a novel determinant, Mar26, in Complex III biogenesis and a physical link between the MICOS and respiratory complexes. The experiments are well-performed and the data are of high quality. However, the functional requirement of MICOS in Rip1-Mar26 mediated complex III biogenesis is not supported by the data shown. In addition, while loss of Mar26 causes a respiratory growth defect, the effects on Complex III assembly are modest and do not necessarily explain a direct functional role of the protein in Complex III biogenesis. At this point, the data are too preliminary to support renaming Fmp10 as Mar26.

Specific points:

1. While the authors convincingly demonstrate that Mic60 can interact with Complex III subunits, the model that this interaction locally mediates Complex III assembly at cristae junctions as a means to pass the diffusion barrier is not supported. The only functional role of MICOS shown by the authors is an extremely subtle and non-quantitative Complex III assembly defect. This defect is found in the absence of both Mic60 and Mic10, which implies it is due to a general consequence of MICOS disruption rather than a specific role of the Mic60-Mar26 interaction. Besides a physical interaction between Mic60 and subunits of Complex III, there are no orthogonal assays that argue that Mar26 promotes Complex III biogenesis in concert with, or in proximity to, MICOS.

We thank the referee for bringing up this important point. In fact, we do not claim that MICOS is strictly required for complex III assembly. We rather suggest that MICOS *facilitates* efficient assembly by recruiting a critical intermediate to a mitochondrial trafficking hub with high protein and lipid sorting activities to assist the spatial and temporal organization of complex III biogenesis. We provide a quantitation of the assembly defects for imported Rip1 and Qcr10 in MICOS-deficient mitochondria demonstrating a reduction down to ~25% for *mic60Δ* mitochondria at early time points (**new** Fig. 6D). However, Qcr8 does not display an assembly defect in MICOS mutants (Fig. EV6C), arguing against a global defect of complex III biogenesis due to membrane architecture alterations. Notably, we show by 2D gel analysis that (in contrast to the steady state situation) the *vast majority of newly imported Rip1* associated with MICOS is part of the small assembly intermediate characterized here (**new** Fig. EV6D). This important experiment reveals that pulldown assays of endogenous protein complexes where Rip1 is almost entirely assembled into mature complex III (Fig. 1C, 6B) poorly reflect the association of newly imported Rip1 with MICOS. We also show that in the absence of Mic10, the Mar26-dependent Rip1 intermediate accumulates at Mic60 to clearly higher levels than in the wild-type situation (**new** Fig. 6E). This finding suggests that while Mic60 as a direct interaction partner supports efficient Rip1 biogenesis, intermediates that rely on this Mic60-facilitated assembly step become trapped in the absence of an intact MICOS. Similarly, we find that the Mar26-dependent Rip1 intermediate interacts with MICOS even in *rho*- mitochondria, i.e. in the absence of ongoing assembly (**new** Fig. EV6D) providing further evidence that this interaction represents a default.

2. A concern is that the MICOS interaction with respiratory complexes, while convincingly shown, may not be specific or physiologically relevant. Controls with other abundant boundary membrane proteins such as Tim complexes and Yme1 should be shown.

As the referee points out, the specificity of the shown interactions is a crucial issue. We now demonstrate that abundant control proteins of both outer and inner mitochondrial membranes and the intermembrane space, including subunits of the F_1F_0 -ATP synthase and the TIM22 complex as well as Yme1, are not co-purified with MICOS in our IgG-based affinity chromatography experiments (Fig. 1A, Fig. 2C, Fig. 6A, **new** Fig. EV1B, Fig. EV1E). Additionally, our stringent SILAC/MS-based interactome assay of Mic60, Cor1 and Mar26 confirm the specificity of the analyzed interactions (high light/heavy peptide ratio counts for specific interactors).

3. While Mar26 expression levels correlate with the presence and absence of a Rip1 assembly intermediate, and loss of Mar26 leads to the subtle accumulation of a Complex III* intermediate, there is no evidence that these minor changes explain the respiratory growth defect of *mar26*-delta cells. As a whole, Complex III assembly looks relatively unaltered in the absence of Mar26. Is Complex III activity specifically altered in the deletion and could this explain the respiratory defect?

We thank the referee for this suggestion. Responding also to reviewer #2, point 1, we have analyzed the enzymatic activities of complex III, IV and CV in *mar26Δ* mitochondria and found that in the absence of Mar26 complex III activity was specifically decreased (**new** Fig.

3D-E, **new** Fig. EV3A). Additionally, we found an increase in ROS production in the absence of Mar26 (**new** Fig. 3F, **new** Fig. EV3B), which may also contribute to the strong growth defect we observe.

4. Mar26 may also indirectly cause the defects in Complex III assembly. While SILAC analysis of Mar26 identifies respiratory complex components, these interactions may not be specific. The authors show accumulation of a small Mar26 complex by blue native PAGE that they claim is the size of Rip1*. Is this dependent on Rip1 or Complex III assembly?

To address this point, we now show by 2D (native / non-native) gel analysis that Mar26 and Rip1 indeed co-migrate in the low molecular weight range on the native dimension (**new** Fig. 4A), and that upon import of radiolabeled Rip1, a small Rip1-containing complex that is present in wild-type and Mar26 overexpressing mitochondria, does not form in the absence of Mar26 (**new** Fig. EV6B). However, formation of the Rip1 intermediate does not depend on the presence of fully assembled complex III (**new** Fig. EV6D).

5. While the authors provide evidence that YBL095W shares minimal homology with Mar26, this does not justify renaming the protein as Mar30. Additionally, it is not clear whether YBL095W contains a transmembrane domain, which may explain the solubility of the protein relative to Mar26 in alkaline extraction assays. Thus, YBL095W may be an intermembrane space localized protein with a completely different functional role.

We concur that too much is still unclear regarding the function of Mrx3 (YBL095W), and therefore refrain from renaming the protein.

6. Can the overexpression of Mar26 be used to provide more functional information on the protein or be used to test their model. For example, does the accumulation of a Rip1* intermediate allow detection of a Rip1*-Mar26 complex? Does it titrate Rip1 from the assembled Complex III or lead to accumulation of Complex III*?

We thank the referee for this excellent suggestion. Indeed, using Mar26 overexpression mitochondria, we now show via 2D (native / non-native) gel analysis that Mar26 and Rip1* precisely co-migrate in the native dimension (**new** Fig. 4A) strongly suggesting complex formation. Upon Mar26 overexpression, Rip1 levels are also elevated, and we did not observe an increased population of complex III lacking Rip1 (Fig. 6A, B). However, in support of the referee's idea that Mar26 should be competing for binding to newly imported Rip1, we now demonstrate that Mar26 overexpression results in a substantial delay of Rip1 assembly into mature respiratory chain supercomplexes (**new** Fig. EV6B).

Minor Points

1. Why do Mic60-ProtA bands in the elution fraction run at a different sizes as compared to Mic60-ProtA in the load fraction? Is the protein A tag cleaved off during the elution?

The assumption of the referee is correct, elution is performed through tobacco etch virus (TEV) cleavage of the protein A tagged bait protein.

2. Mic12 -ProtA western blots should be shown in Figure 1A.

We agree that this would be optimal, but unfortunately the detection of this particular antibody failed in this case. However, the co-isolation efficiency for other MICOS components (Mic60, Mic10 and Mic27) is very similar for the bait proteins Mic60-ProtA and Mic12-ProtA, showing that the pulldown experiment worked equally well for Mic12-ProtA.

3. Typographical error on Page 13, Line13 - "und" should be "and".

4. Typographical error on the top of page 15 - reference to Fig. S4C should be S4B

We thank the referee for pointing out these errors, which we have resolved in the revised manuscript.

Referee #2:

In this work, Zerbes et al characterize the mitochondrial protein Fmp10 (which they now refer to Mar26) and suggest that it plays an important role in late stages in the biogenesis of respiratory complex III. The protein also co-purifies with the MICOS complex, which is located at the junction between the inner boundary membrane and the cristae, invaginations of the inner membrane. Based on this, the authors suggest that late stages complex III biogenesis at this site could be a possibility to determine the distribution of respiratory chain complexes between the different compartments of the inner membrane. Overall, the technical quality of the experiments is high and the manuscript is well written.

We thank the referee for this very positive assessment of our study.

However, a few key experiments are missing, limiting the molecular insights into the function of Mar26 and questioning the overall significance of the study. Hence, the authors need to experimentally address a number of key questions, the results of which could potentially require a substantial revision of the text.

Major points:

1. The exact molecular function of Mar26 is yet poorly defined. While absence of Mar26 provokes a massive growth defect on respiratory medium (Fig 5A), the accumulation of respiratory chain enzymes including complex III is not affected (Fig 5C and D). This would suggest that Mar26 does not play an important role in complex III biogenesis, but rather in something else. In order to substantiate that absence of Mar26 provokes a specific defect in complex III biogenesis, the authors should provide data showing the specific enzymes activities of oxidative phosphorylation complexes from WT and mar26D mitochondria.

Specifically, the authors should determine quinol-dependent cytochrome c reductase activity for complex III, KCN-sensitive cytochrome c oxidation for complex IV and oligomycin-sensitive ATPase activity. If these experiments reveal that complex III activity is unchanged or not substantially reduced, the conclusion could be that the slight changes in Rip1 biogenesis in absence of Mar26 could be a secondary consequence of Mar26 loss, requiring then the identification of the primary function of Mar26 in mitochondria.

As indicated also in our response to referee 1, we have followed up this excellent suggestion and analyzed the individual activities of complex III, IV and V in *mar26Δ* mitochondria. We found that complex III activity was specifically reduced in the absence of Mar26, but not the activities of complex IV or V (**new** Fig. 3D-E, **new** Fig. EV3A). Additionally, we found an increase in ROS production in the absence of Mar26 (**new** Fig. 3F, **new** Fig. EV3B) which might also contribute to the strong growth defect we observe.

Furthermore, we have included additional lines of evidence firmly linking the role of Mar26 to complex III biogenesis. Mar26 co-migrates with Rip1 in the low molecular weight range in 2D gel analysis (**new** Fig. 4A) in agreement with our previous finding that the small Rip1 intermediate is Mar26 dependent (Fig. 3G), a point we now show also for newly imported Rip1 (**new** Fig. EV6B). Importantly, we demonstrate that in mitochondria lacking Rip1 or its biogenesis factors Bcs1 or Mzm1, Mar26 accumulates at late complex III assembly intermediates with enhanced efficiency (**new** Fig. 4B, 4D, EV4). Moreover, Mar26 co-isolates immature complex III species in *rip1Δ* mitochondria (**new** Fig. 4C). These results clearly establish that Mar26 plays a direct role in complex III biogenesis at the step of Rip1 assembly.

2. The authors suggest that Mar26 plays a role in Rip1 incorporation into complex III. Their most significant result is that a small complex of Rip1 is absent in BN-PAGE in the mutant lacking Mar26 (Fig 5D). The composition of this minor Rip1 complex is yet unknown, but it is possible that Mar26 is one of the components. Hence it would be important to isolate Mar26 complexes and determine their composition by MS and through antibodies. Moreover, there is a whole literature on Rip1 biogenesis, where the protein is first completely imported into the matrix, where it receives the Fe-S cluster, which is stabilized by Mzm1. Next, the mature Rip1 is translocated by the ATPase Bcs1, which associates with the 500 kDa intermediate to facilitate Rip1 assembly into this complex. Data from this study shows that incorporation of newly imported Rip1 into the complex is accelerated upon deletion of Mar26 (Fig 6B), suggesting that presence of Mar26 might be inhibiting assembly of newly imported Rip1. Hence, the exact role of Mar26 is yet ill-defined. The authors should test whether Mar26 directly and specifically interacts with newly imported Rip1 and test whether Rip1 would require the residues necessary for coordination of the Fe-S cluster for this interaction. Likewise, it would be important to elucidate the relationship of Mar26 with the known Rip1 biogenesis factors Mzm1 and Bcs1. Do they collaborate in Rip1 biogenesis, or do they work on alternative pathways?

We thank the referee for bringing up these important issues. In addition to the evidence mentioned above linking Mar26 to complex III assembly, we now show that Mar26 specifically interacts with newly imported Rip1 in wild-type mitochondria, but not in the

absence of Bcs1 (**new** Fig. 5C). Importantly, the co-isolation efficiency in wild-type is much higher for newly imported than for endogenous Rip1. These results indicate that Mar26 indeed selectively interacts with newly imported, unassembled Rip1 and that the interaction takes place after Rip1 translocation from the matrix into the inner membrane with the active site facing the intermembrane space.

Moreover, we now demonstrate that variants of Rip1 with disrupted Fe-S cluster binding (S183C) or lacking a structurally important disulfide bond (C164Y/C180L) lose their ability to interact with Mar26 (**new** Fig. 5D), confirming that Mar26 interacts with membrane-inserted, cofactor-containing assembly-competent Rip1 molecules on the intermembrane space side of the inner membrane.

The requested mass spectrometry analysis of the Mar26 interactome was already included in our study (Table S1) and fully supports our conclusions regarding its interactions with both the respiratory chain and MICOS.

3. The authors suggest that the MICOS complex, through interaction with assembly factors like Mar26, facilitates respiratory complex assembly specifically to cristae junctions, likely to distribute fully assembled complexes into cristae membranes. While this is an appealing hypothesis, this work does not provide substantial evidence that this is the case. If MICOS would play a prominent role in organizing late-stage complex III assembly, one would expect that MICOS' interactions with assembly factors and assembly intermediates would be rather quantitative. However, data presented in Fig 1 and 2 show that MICOS primarily interacts with fully assembled complex III (found in supercomplexes) and not with the 500 kDa intermediate of complex III assembly. Likewise, it is not known how much of Mar26 and the small Rip1 intermediate is bound to MICOS. Data presented in Fig S4C shows that Mar26 forms several separate complexes. The authors should determine the migration pattern of Mar26 and MICOS complexes from comparative co-migration analyses using BN-PAGE and test how Mar26 complexes change in complex III assembly or MICOS mutants.

We have followed up these very helpful suggestions and now include, as outlined above, substantial new evidence for Mar26 binding to partially assembled complex III species in Rip1 assembly mutants or in *rip1* Δ mitochondria (**new** Fig. 4B-D, **new** Fig. EV4), confirming its role in complex III assembly. We also demonstrate that not only Mar26, but also MICOS interacts with newly imported Rip1 with high preference (**new** Fig. 5C, **new** Fig. EV6D): In contrast to endogenous Rip1, most of which is assembled in mature respiratory chain complexes, the vast majority of co-isolated newly imported Rip1 is part of the small intermediate (**new** Fig. EV6D).

Additionally, in line with the reviewer's expectations, we show that the co-isolation efficiency of Mar26 with Mic60-ProtA is significantly greater than that of respiratory chain complex subunits such as Cyt1, Qcr8, Cox2 or Cox4 (Fig. 6A). Moreover, in MICOS pulldown eluates the ratio of Rip1* intermediate to mature respiratory chain supercomplexes is much higher than in solubilized mitochondria (Fig. 6B). Taken together, our results demonstrate that MICOS interacts preferentially with Mar26 and the Rip1* intermediate compared to mature respiratory chain complexes.

Regarding the role of MICOS in complex III biogenesis, our results establish that MICOS enhances the efficiency of assembly, likely by localizing the Rip1* intermediate to crista

junctions. Supporting this conclusion, we now show that in the absence of Mic10, the Mar26-dependent Rip1 intermediate accumulates at Mic60 (**new Fig. 6E**). This result indicates that while Mic60 as a direct interaction partner is required for efficient Rip1 biogenesis, intermediates that rely on Mic60-facilitated assembly can become trapped there in the absence of an intact MICOS. Similarly, we find that the Mar26-dependent Rip1 intermediate interacts with MICOS even in *rho*- mitochondria, i.e. in the absence of ongoing assembly (**new Fig. EV6D**).

Referee #3:

The study by Zerbes et al. addresses the interplay between cristae organization and assembly of respiratory (super-)complexes. The authors show that the MICOS complex is in physical contact to respiratory supercomplexes and they identify a novel player, Mar26 (and also its homolog Mar30), which specifically helps to maintain an assembly-competent Rip1 pool which appears to promote supercomplex assembly in the local neighbourhood of the MICOS complex at crista junctions. Moreover, the authors show that deletion of MICOS subunits alters the assembly kinetics of distinct subunits of supercomplexes in a specific manner. Overall, I think that this is an excellent and very important study which provides fundamental insights into how complex assembly is mechanistically linked to cristae organization. There is indeed really very little known in this regard. It reveals a specific novel pathway for a late-assembly step in complex III assembly linked to the MICOS complex. Moreover, the data shown is very convincing and very well described. I really only have minor suggestions that might improve the study further.

We thank the reviewer for the very positive evaluation of our study.

1. The assembly kinetics of Rip1 into complex was proposed to be impaired in strains lacking Mic10 or Mic60. This is very interesting and could indeed be true. Yet, it is not fully convincing and should be quantified (see Fig. 7C).

We have included quantitation of our Rip1 and Qcr10 assembly assays with wild-type and MICOS-deficient mutant mitochondria showing a substantial decrease in assembly down to ~25% in the absence of MICOS core subunits (**new Fig. 6D**).

2. Fig. 1C. (lanes 23/24/29/30). The Rip1-containing complex at 140 kDa in the elution fraction is difficult to see, even in the long exposure. This could be improved (compare Fig. 7B lane 2). In addition, I suggest to decorate also the blot of the BN-PAGE (Fig. 7B with *mar26-delta* and *mar26-OE*) with the Mar26-antibody testing whether Mar26 also has a native size of around 140 kDa. This would strengthen the interpretation that Mar26 and Rip1 are forming an assembly-competent intermediate.

We have replaced the Rip1 blot (long exposure) in Fig. 1C with an even longer exposure to show the Rip1 intermediate more clearly. Additionally, we now include a 2D (native / non-native) gel analysis demonstrating that Mar26 and Rip1 precisely co-migrate in the low

molecular weight region of the native dimension (**new** Fig. 4A) supporting our conclusion that they form a complex.

3. Fig. 2A. Related to this also a long exposure of Rip1 in Fig. 2A would be helpful to see whether deletion of Mic10 has an effect on the 140 kDa intermediate.

Indeed, we find that in the absence of Mic10, the Rip1 intermediate accumulates at Mic60 (**new** Fig. 6E), lending support to our conclusion that MICOS binds this intermediate to facilitate its incorporation into complex III, resulting in trapping of the intermediate at Mic60 in the absence of Mic10.

4. Fig. 4B. Here Cyt1 is shown. What about the effect of loss of Mar26 on the interaction of Mic60 with with Rip1 or Cox1?

This is an important point. Our Mic60 pulldown analysis shows that co-isolation of Rip1 is decreased in the *mar26* Δ background, whereas co-isolation of other subunits of complex III (Cyt1, Qcr8) or complex IV (Cox2, Cox4) is unaffected (Fig. 6A, lanes 17,18). These results indicate that Rip1 co-isolated in the WT background reflects a combination of Rip1 in mature respiratory chain complexes and in the Rip1* intermediate. Importantly, the relative amounts of Rip1 in the intermediate vs. mature respiratory chain (super-)complexes are very clearly shifted in the MICOS pulldown eluates, showing a substantially higher co-isolation efficiency for unassembled Rip1 (Fig. 6B). We now additionally demonstrate that for newly imported Rip1 co-isolated with MICOS the proportion of Rip1 in the intermediate is vastly larger than that of fully assembled Rip1 (**new** Fig. EV6D), showing that MICOS preferentially interacts with newly imported Rip1 as part of its Mar26 intermediate.

5. The homologs of Mar26 in animals are mitochondrial thioesterases. Do the authors think that this activity and its link to fatty acid metabolism is relevant. I suggest to discuss this aspect in the discussion section.

We agree that this point should be addressed and have included a discussion of this issue as requested.

Dear Heike,
Dear Martin,

Thank you for submitting your revised manuscript. It has now been seen by all of the original referees and I performed the necessary editorial checks. My apologies for the delay in getting back to you, which is due to the recent conference travels.

As you can see, the referees find that the study is significantly improved during revision and recommend publication. However, I need you to address the editorial points below before I can accept the manuscript.

- The maximum number of keywords we can accommodate is 5. Since there are currently 6 keywords, please remove one of the keywords.
- Please remove the 'Author contributions' section from the manuscript.
- As per our format requirements, in the reference list, citations should be listed in alphabetical order and then chronologically, with the authors' surnames and initials inverted; where there are more than 10 authors on a paper, 10 will be listed, followed by 'et al.'. Please see <https://www.embopress.org/page/journal/14693178/authorguide#referencesformat>
- We note that Figure 5C is currently not called out in the text.
- "(Supplemental) Table EV1" is a dataset so it needs to be uploaded and renamed as such - Dataset EV1. Manuscript callouts need to be updated accordingly. Table EV2 and Table EV3 are provided in the manuscript file, which need to be removed and uploaded as separate files but renamed to Table EV1 and EV2 - manuscript callouts included.
- During our routine image analysis, our data integrity analyst Christopher Rickerby noted some irregularities in Figure EV1. Therefore, please provide source data for this figure as well.
- We noted that some portions of the blots in the source data are redacted. We are of the opinion that this may be due to possible plans to publish those portions separately. Please clarify.
- Please resubmit the source data as one folder per figure. Also, please fill the source data checklist, which was sent by our Source Data Coordinator Dr. Hannah Sonntag on 01.05.2024, also attached to this letter.
- Expanded view information section should be renamed as "Expanded View Figure Legends".
- The manuscript sections should be in the following order: Title page - Abstract & Keywords - Introduction - Results - Discussion - Methods - Data Availability - Acknowledgments - Disclosure Statement & Competing Interests - References - Figure Legends - (Main Tables with legends) - Expanded View Figure Legends.
- Our production/data editors have asked you to clarify several points in the figure legends:
 - Please note that the figure EV 2d is missing in the manuscript, however the legend for the same is provided. This needs to be rectified.
 - Please define the annotated p values **** in the legend of figure 3d, f; as appropriate.
 - Please note that the exact p values are not provided in the legends of figures 3d, f.
 - Please indicate the statistical test used for data analysis in the legend of figure 2a.
 - Although 'n' is provided, please describe the nature of entity for 'n' in the legends of figures 3d-f; 6d; EV 3b.
 - Please note that the measure of center for the error bars needs to be defined in the legend of figure 6d.
 - Please note that the black circles are not defined in the legend of figure EV 6b. This needs to be rectified.
- Papers published in EMBO Reports include a 'synopsis' and 'bullet points' to further enhance discoverability. Both are displayed on the html version of the paper and are freely accessible to all readers. The synopsis includes a short standfirst summarizing the study in 1 or 2 sentences (max 35 words) that summarize the paper and are provided by the authors and streamlined by the handling editor. I would therefore ask you to include your synopsis blurb and 3-5 bullet points listing the key experimental findings.
- In addition, please provide an image for the synopsis. This image should provide a rapid overview of the question addressed in the study but still needs to be kept fairly modest since the image size cannot exceed 550 (width) x 300-600 (height) pixels.

Thank you again for giving us to consider your manuscript for EMBO Reports, I look forward to your minor revision.

Kind regards,

Deniz

--

Deniz Senyilmaz Tiebe, PhD
Senior Scientific Editor
EMBO Reports

Referee #1:

The authors have addressed very well the points this reviewers has raised on the initially submitted version. Specifically, they now provide in this revised version further evidence for the role of Mar26 in the assembly of Rip1 and the connection between

this last step of complex III assembly and the MICOS complex. This work will be of great interest to the community working on mitochondrial biogenesis.

Referee #2:

The revised manuscript from Zerbes et al is substantially improved and the authors have addressed the concerns raised in my original reviews. I recommend publication of this interesting study.

Referee #3:

The authors have fully addressed all my concerns. Nice work.

*EMBOR-2024-59422V2 revised*Response to editorial points

Thank you again for the very competent and careful handling of our manuscript and for the clear communication on points that we can still improve.

- The maximum number of keywords we can accommodate is 5. Since there are currently 6 keywords, please remove one of the keywords.
- Please remove the 'Author contributions' section from the manuscript.

We have removed one keyword as well as the section Author contributions.

- As per our format requirements, in the reference list, citations should be listed in alphabetical order and then chronologically, with the authors' surnames and initials inverted; where there are more than 10 authors on a paper, 10 will be listed, followed by 'et al.'. Please see <https://www.embopress.org/page/journal/14693178/authorguide#referencesformat>

We have limited the number of listed authors per reference to 10 throughout the reference list.

- We note that Figure 5C is currently not called out in the text.

Thank you for catching this, we had described the findings without mentioning the panel. This has been corrected.

- "(Supplemental) Table EV1" is a dataset so it needs to be uploaded and renamed as such - Dataset EV1. Manuscript callouts need to be updated accordingly. Table EV2 and Table EV3 are provided in the manuscript file, which need to be removed and uploaded as separate files but renamed to Table EV1 and EV2 - manuscript callouts included.

We have changed the naming of the EV data accordingly and externalized the two tables.

- During our routine image analysis, our data integrity analyst Christopher Rickerby noted some irregularities in Figure EV1. Therefore, please provide source data for this figure as well.

We include the original data for Figure EV1 and hope that any issues can be resolved now. Due to technical problems with the original source file for Fig. EV1E, we have replaced that experiment with an equivalent repeat experiment. All scientific conclusions are unaffected by this exchange.

- We noted that some portions of the blots in the source data are redacted. We are of the opinion that this may be due to possible plans to publish those portions separately. Please clarify.

Indeed, we removed original data that go beyond what we show in the manuscript to enable us to publish these data separately. We hope for your understanding in this regard.

- Please resubmit the source data as one folder per figure. Also, please fill the source data checklist, which was sent by our Source Data Coordinator Dr. Hannah Sonntag on 01.05.2024, also attached to this letter.

We have restructured the source data accordingly and have filled the source data checklist. In addition to the source data submitted with the revised version for the main figures, we include the source data for Figure EV1, as requested.

- Expanded view information section should be renamed as "Expanded View Figure Legends".
- The manuscript sections should be in the following order: Title page - Abstract & Keywords - Introduction - Results - Discussion - Methods - Data Availability - Acknowledgments - Disclosure Statement & Competing Interests - References - Figure Legends - (Main Tables with legends) - Expanded View Figure Legends.

We have made the requested changes.

- Our production/data editors have asked you to clarify several points in the figure legends:
 - Please note that the figure EV 2d is missing in the manuscript, however the legend for the same is provided. This needs to be rectified.
 - Please define the annotated p values **** in the legend of figure 3d, f; as appropriate.
 - Please note that the exact p values are not provided in the legends of figures 3d, f.
 - Please indicate the statistical test used for data analysis in the legend of figure 2a.
 - Although 'n' is provided, please describe the nature of entity for 'n' in the legends of figures 3d-f; 6d; EV 3b.
 - Please note that the measure of center for the error bars needs to be defined in the legend of figure 6d.
 - Please note that the black circles are not defined in the legend of figure EV 6b. This needs to be rectified.

We have added the requested information to the figure legends and removed the legend for Figure EV2E which was a duplicate following the moving of a figure panel to Figure EV3.

- Papers published in EMBO Reports include a 'synopsis' and 'bullet points' to further enhance discoverability. Both are displayed on the html version of the paper and are freely accessible to all readers. The synopsis includes a short standfirst summarizing the study in 1 or 2 sentences (max 35 words) that summarize the paper and are provided by the authors and streamlined by the handling editor. I would therefore ask you to include your synopsis blurb and 3-5 bullet points listing the key experimental findings.

We now include a synopsis as well as bullet points for our manuscript.

- In addition, please provide an image for the synopsis. This image should provide a rapid overview of the question addressed in the study but still needs to be kept fairly modest since the image size cannot exceed 550 (width) x 300-600 (height) pixels.

We have prepared a summary image for our manuscript that we now include in our submission.

Dear Heike,
Dear Martin,

Thank you for submitting your revised manuscript and providing the additional files. I have now looked at everything and all is fine. Therefore, I am very pleased to accept your manuscript for publication in EMBO Reports.

Congratulations on a nice work!

Kind regards,

Deniz
--
Deniz Senyilmaz Tiebe, PhD
Senior Scientific Editor
EMBO Reports
